

# Effects of snow grain shape on climate simulations: Sensitivity tests with the Norwegian Earth System Model

Petri Räisänen[1], Risto Makkonen[2], Alf Kirkevåg[3], and Jens Boldingh Debernard[3]

[1]Finnish Meteorological Institute, P.O. Box 503, FI-00101 Helsinki, Finland
[2]Dept. of Physics, University of Helsinki, P.O. Box 64, FI-00014 University of Helsinki, Finland
[3]Norwegian Meteorological Institute, P.O. Box 43, Blindern, N-0313 Oslo, Norway

*Correspondence to:* Petri Räisänen (petri.raisanen@fmi.fi)

**Abstract.** Snow consists of non-spherical grains of various shapes and sizes. Still, in radiative transfer calculations, snow grains are often treated as spherical. This also applies to the computation of snow albedo in the Snow, Ice, and Aerosol Radiation (SNICAR) model and in the Los Alamos sea ice model, version 4 (CICE4), both of which are employed in the Community Earth System Model and in the Norwegian Earth System Model (NorESM). In this study, we evaluate the effect of snow grain shape on climate simulated by NorESM in a slab ocean configuration of the model. An experiment with spherical snow grains (SPH) is compared with another (NONSPH) in which the snow shortwave single-scattering properties are based on a combination of three non-spherical snow grain shapes, optimized using measurements of angular scattering by blowing snow. The key difference between these treatments is that the asymmetry parameter is smaller in the non-spherical case (0.77–0.78 in the visible region) than in the spherical case ($\approx 0.89$). Therefore, for a given snow grain size, the use of non-spherical snow grains leads to a higher snow broadband albedo, typically by 0.02–0.03. Considering the spherical case as the baseline, this results in an instantaneous negative change in net shortwave radiation with a global-mean top-of-the-model value of ca. $-0.22 \, \mathrm{W \, m^{-2}}$. Although this global-mean radiative effect is rather modest, the impacts on the climate simulated by NorESM are substantial. The global annual-mean 2-m air temperature in NONSPH is $1.17 \, \mathrm{K}$ lower than in SPH, with substantially larger differences at high latitudes. The climatic response is amplified by strong snow and sea ice feedbacks. It is further demonstrated that the effect of snow grain shape could be largely offset by adjusting the snow grain size. When assuming non-spherical snow grains with the parameterized grain size increased by ca. 70%, the climatic differences to the SPH experiment become very small. Finally, the impact of assumed snow grain shape on the radiative effects of absorbing aerosols in snow is discussed.

## 1 Introduction

Snow albedo, defined as the fraction of incoming solar energy reflected upwards by the snow surface, plays a fundamental role in the surface energy budget of snow-covered regions. The high albedo of snow contributes to the cold climate of high latitudes. Snow albedo is associated with a major positive feedback mechanism, the snow-albedo feedback. Decreasing snow cover on land and sea ice acts to reduce the surface albedo, thereby increasing the solar radiation absorbed by the underlying surface and accelerating the warming, both on the seasonal time scale (i.e., the snow melt in spring) and in the context of climate change (e.g., Hall and Qu, 2006; Fletcher et al., 2012).



The treatment of snow albedo varies considerably among climate models (e.g., Wang and Zeng, 2010). Many of the albedo schemes are semiempirical rather than based on radiative transfer modelling. For example, a simple scheme that diagnoses snow broadband albedo as a function of temperature was used in the ECHAM5 model (Roeckner et al., 2003). In ECHAM6 (Giorgetta et al., 2013), it has been replaced with a more comprehensive parameterization originally adapted from the Biosphere-Atmosphere Transfer Scheme (BATS) (Dickinson et al., 1986), which distinguishes between snow albedo for visible and

near-infrared radiation as well as between direct and diffuse solar radiation, and also considers snow aging and impurities. Still, snow albedo is computed without explicitly considering the size and shape of snow grains. Albedo schemes that link the snow albedo to snow grain size through radiative transfer modeling also exist (Flanner and Zender, 2005; Gardner and Sharp, 2010; Aoki et al., 2011). To the best of our knowledge, spherical snow grains are assumed in all such schemes.

It is, however, well known that snow grains are non-spherical and often irregular in shape. Furthermore, the single-scattering

properties (SSPs) of non-spherical particles can differ greatly from those of spheres. While there is no concensus on which shape model should be used to represent snow grains, several alternatives have been considered, including Koch fractals (Kokhanovsky and Zege, 2004), cylindrical particles or prolate ellipsoids with rough surfaces (Tanikawa et al., 2006), aggregates of columns (Jin et al., 2008), and a mixture of columns and plates with rough surfaces (Zege et al., 2011). Liou et al. (2014) suggested the use of hexagonal plates/columns and Koch snowflakes for new snow and spheroids for old snow. Räisä-

nen et al. (2015) developed a parameterization for the SSPs of non-spherical snow grains (the single-scattering co-albedo $\beta$, the asymmetry parameter $g$ and the phase function $P_{11}$) based on a combination of three habits, a so-called optimized habit combination (OHC): severely rough (SR) droxtals, aggregates of SR plates and strongly distorted Koch fractals. This choice was based on fitting the phase function to measurements of angular scattering by blowing snow. Most recently, Dang et al. (2016) evaluated the effects of snow grain non-sphericity on snow albedo using a parameterization of ice crystal asymmetry

parameter developed for hexagonal prisms with rough surfaces (Fu, 2007).

A related uncertainty factor is the treatment of snow grain size. For radiative transfer, the most relevant measure of the size of non-spherical particles (including snow grains) is the volume-to-projected area equivalent effective radius (e.g., Mitchell, 2002),

$$r_e = 0.75V/P, \qquad (1)$$

where $V$ is the total volume and $P$ the total projected area of the particle population. For convex particles such as spheres, $r_e$ can be derived from the snow surface specific area, which can be measured in laboratory with techniques such as methane adsorption (e.g., Legagneux et al., 2002), X-ray microtomograhy (e.g., Flin et al., 2005) and stereology (e.g., Matzl and Schneebeli, 2010). These techniques are, however, not applicable to extensive monitoring of snow grain size. Remote sensing algorithms for retrieving snow grain $r_e$ have been developed (e.g., Lyapustin et al., 2009; Kokhanovsky et al., 2011; Zege et al., 2011) but

these algorithms are, to some extent, sensitive to assumptions about snow grain shape, which makes the uncertainties in grain size and shape intertwined. To our knowledge, no well-established climatology of snow grain size exists at present.

The issues of grain shape and size are also intertwined from the point of view of snow albedo. It has been shown that the measured spectral albedo of snow can be fitted by radiative transfer calculations under the assumption of spherical snow grains



(Wiscombe and Warren, 1980; Grenfell et al., 1994; Aoki et al., 2000), when the effective snow grain size is considered an

adjustable parameter (i.e., determined based on albedo rather than microphysical measurements). Recently, Dang et al. (2016) demonstrated that the spectral albedo of a snowpack consisting of non-spherical snow grains can indeed be mimicked by using smaller grains with spherical shape. They found that the scaling factor between the effective radius of non-spherical and spherical grains producing the same albedo depends on the aspect ratio of the non-spherical grains, peaking at 2.4 for equidimensional snow grains. If, instead, the same size is assumed for spheres and equidimensional hexagonal snow grains,

the broadband albedo of an optically thick snowpack is higher for the latter by ca. 0.03–0.05. The physical reason for this is that the asymmetry parameter is lower for hexagonal snow grains, ca. 0.74 in the visible region in the equidimensional case, as compared with ca. 0.89 for spheres, owing to the larger sideward scattering by hexagonal grains. For comparison, for the OHC in Räisänen et al. (2015), $g \approx 0.77$–$0.78$ in the visible region.

    The primary research question addressed in this article is: how are climate model simulations influenced by the choice of

snow grain shape in the snow albedo calculation? A slab ocean configuration of the Norwegian Earth System Model NorESM (Bentsen et al., 2013; Iversen et al., 2013; Kirkevåg et al., 2013; Tjiputra et al., 2013) is employed for the experiments. NorESM is very well suited for this study because snow albedo is computed using radiative transfer modelling (two-stream approximations) based on the physical properties of the snow layer: the snow water equivalent (SWE), the effective radius $r_e$ of snow grains, and concentrations of absorbing aerosols in snow. The $r_e$ of snow grains is derived prognostically over land

(Flanner and Zender, 2005, 2006), while over sea ice, it is parameterized as a function of temperature. The SSPs of snow grains needed for the albedo calculation are computed as a function of $r_e$, assuming spherical snow grains in the default version of the model.

    In the current work, the default spherical shape assumption is compared with the OHC assumption of Räisänen et al. (2015). To the best of our knowledge, this is the first time that non-spherical snow grains have been considered in climate model

simulations. Given the uncertainties related to the representation of snow grain size and shape, this work is best viewed as a sensitivity study. Indeed, it is found that changing the snow grain shape has a substantial impact on the simulated climate especially at high latitudes, due to the strong snow and sea ice feedbacks induced.

    The outline of this paper is as follows. First, in Sect. 2, the NorESM model is introduced, with most focus on the parts relevant for the calculation of snow albedo. The model experiments are briefly introduced in Sect. 3. The results are reported in

Sect. 4, which is divided into three subsections, focusing on the instantaneous radiative effect ("radiative forcing") of changed snow grain shape (Sect. 4.1), on the temperature and precipitation response (Sect. 4.2), and on snow cover, sea ice and albedo (Sect. 4.3). Section 5 then revisits the competing effects of snow grain shape and size, demonstrating that (at least from the technical point of view) the climatic effects induced by introducing the non-spherical shape assumption could be largely offset by retuning the snow grain size by ca. 70%. After that, the impact of snow grain shape assumptions on aerosol absorption in

snow is discussed in Sect. 6. Some of the main results and key uncertainties associated with this work are further discussed in Sect. 7, and a summary is provided in Sect. 8.



## 2 Model

The experiments discussed in this paper were conducted with the Norwegian Earth System Model (NorESM) version 1. NorESM is largely based on the Community Climate System Model, version 4 (CCSM4) (Gent et al., 2011; Vertenstein et al., 2010) and the Community Earth System model, version 1 (CESM1) (Hurrell et al., 2013), with modified atmosphere and ocean components. The actual model model code employed here is based on the version of NorESM with interactive carbon cycle (Tjiputra et al., 2013), but in the current experiments, atmospheric $CO_2$ is prescribed and a slab ocean model is used instead of the ocean general circulation and carbon cycle components. The description of the physical climate system follows NorESM1-M, which is detailed in Bentsen et al. (2013), Iversen et al. (2013) and Kirkevåg et al. (2013).

In the present work, the following model components are employed:

- The Oslo version of the Community Atmosphere Model (CAM4-Oslo), which differs from the original CAM4 (Neale et al., 2010, 2013) through modified chemistry-aerosol-cloud-radiation interaction schemes (Kirkevåg et al., 2013). The finite-volume dynamic core is employed. The horizontal resolution is is 1.9°in longitude and 2.5°in latitude, with 26 levels in the vertical and the model top at 2.19 hPa.

- The CCSM4 slab ocean model (Bitz et al., 2012). The $Q$-flux (representing the implied horizontal and vertical heat flux into/out of the local mixed-layer column) and the spatially (but not temporally) varying depth of the mixed layer are based on the CMIP5 preindustrial control simulation with the fully coupled version of NorESM1-M (Bentsen et al., 2013). This choice has some implications for the current experiments, which will be noted in Sects. 4.2 and 4.3.

- The version of the Los Alamos sea ice model CICE4 used in CCSM4 (Gent et al., 2011; Holland et al., 2012).

- The original version 4 of the Community Land Model (CLM4) of CCSM4 (Oleson et al., 2010; Lawrence et al., 2011), which includes the SNow, ICe and Aerosol Radiative model (SNICAR; Flanner and Zender, 2005, 2006).

- The CCSM4 coupler CPL7 (Craig et al., 2012).

For this study, the most relevant part of the model physics is the computation of surface albedo in the presence of snow. The surface albedo calculation over land accounts for the effects of snow, canopy and the underlying ground (Oleson et al., 2010), while over sea ice, the contributions by snow-covered ice, bare ice and melt ponds are considered (Holland et al., 2012). Both over land and sea ice, the reflection and absorption of solar radiation by snow are computed using a two-stream approximation for multiple scattering (Flanner and Zender, 2005; Flanner et al., 2007; Briegleb and Light, 2007). In SNICAR, the delta-Eddington approximation (Joseph et al., 1976) is applied in the visible region and the delta-hemispheric mean approximation (Toon et al., 1989) in the near-infrared bands, while in CICE4, the delta-Eddington approximation is used for all bands. As input data, the two-stream calculations require, for each spectral band, the albedo of the underlying surface, the solar zenith angle, and the optical thickness $\tau$, the single-scattering albedo $\omega$ (or equivalently the co-albedo $\beta = 1 - \omega$) and the asymmetry parameter $g$ for each snow layer (up to five layers in SNICAR, only one in CICE4). These parameters are obtained by adding the contributions of pure snow and absorbing aerosols in snow (hydrophobic and hydrophilic black carbon and mineral dust).





For pure snow, $\tau$ depends both on the snow water equivalent (SWE) and the effective radius ($r_e$) of snow grains, while $\omega$ and
$g$ are functions of $r_e$ only.

For further use in Sect. 5, we discuss here the computation of snow grain $r_e$ in some detail. In CICE4, $r_e$ is diagnosed
as a function of snow surface temperature $T_s$ such that $r_e$ varies between a value of $r_{e,\text{nonmelt}}$ for $T_s \leq -1.5^\circ C$ and $r_{e,\text{melt}}$
for $T_s \geq 0^\circ C$, with a linear interpolation in-between. The default parameter values in NorESM are $r_{e,\text{nonmelt}} = 500\,\mu\text{m}$ and
$r_{e,\text{melt}} = 1500\,\mu\text{m}$. In contrast, in SNICAR $r_e$ is a prognostic variable simulated with a snow aging routine for each snow layer
separately. The equation for $r_e$ at time step $t$ is (Flanner and Zender, 2006; Oleson et al., 2010)

$$r_e(t) = \left[r_e(t-1) + dr_{e,\text{dry}} + dr_{e,\text{wet}}\right] f_{\text{old}} + r_{e,0} f_{\text{new}} + r_{e,\text{rfrz}} f_{\text{rfrz}}. \tag{2}$$

Here, $r_e(t-1)$ is $r_e$ at timestep $t-1$, $f_{\text{old}}$, $f_{\text{new}}$ and $f_{\text{rfrz}}$ are the fractions of old snow (i.e., snow carrying over from the
previous time step), freshly-fallen snow, and refrozen liquid water, respectively, $r_{e,0} = 54.5\,\mu\text{m}$, and $r_{e,\text{rfrz}} = 1000\,\mu\text{m}$. The
rate of change of $r_e$ due to dry snow metamorphism is

$$\frac{dr_{e,\text{dry}}}{dt} = \text{xdrdt} \times \left(\frac{dr_e}{dt}\right)_0 \left(\frac{\eta}{(r_e - r_{e,0}) + \eta}\right)^{1/\kappa}, \tag{3}$$

where xdrdt is an arbitrary tuning constant (by default, xdrdt $= 1$) and the parameters $(dr_e/dt)_0$, $\eta$ and $\kappa$ are retrieved from a
lookup table as a function of snow temperature, vertical temperature gradient in snow, and snow density (Flanner and Zender,
2006). Furthermore, the rate of change of $r_e$ due to wet snow metamorphism is

$$\frac{dr_{e,\text{wet}}}{dt} = \text{xdrdt} \times \frac{C_1 f_{\text{liq}}^3}{4\pi r_e^2}, \tag{4}$$

where $C_1 = 4.22 \cdot 10^5\,\mu\text{m}^3\,\text{s}^{-1}$ and $f_{\text{liq}}$ is the mass fraction of liquid water in the snowpack (Brun, 1989). The values of $r_e$ are
allowed to vary between a minimum of $r_{e,0}$ and a maximum of $r_{e,\text{max}} = 1500\,\mu\text{m}$.

## 2.1 Snow single-scattering properties

In both SNICAR and CICE, the SSPs of snow are tabulated as a function of snow grain $r_e$, for five spectral intervals in SNICAR
(0.2–0.7, 0.7–1.0, 1.0–1.2, 1.2–1.5 and 1.5–5.0 $\mu$m) and three in CICE (0.2–0.7, 0.7–1.19, and 1.19–5.0 $\mu$m). Here, two options
are considered for the snow grain shape: spherical snow grains and the OHC introduced in Räisänen et al. (2015). In each case,
the SSPs were derived as detailed in Appendix A. The differences in extinction coefficient between spheres and the OHC are
very minor (not shown), but the differences in asymmetry parameter and single-scattering co-albedo are more substantial, as
shown in Fig. 1a–b for the spectral bands in SNICAR. The asymmetry parameter is considerably smaller for the OHC than
for spheres, with values of $g \sim 0.78$ for the OHC and $g \sim 0.89$ for spheres in the three most weakly absorbing bands (0.2–0.7,
0.7–1.0, and 1.0–1.2 $\mu$m). Thus there is more sideward and backward scattering in snow for the OHC than for spheres, which
tends to make snow albedo higher. The effect of smaller $g$ is partially counteracted by $\beta$ being higher (i.e., $\omega$ being lower) for
the OHC than for spheres in the first four spectral bands, by 12–30% depending on $r_e$ and the band. This means that the relative
probability of absorption as compared to scattering is enhanced. The net effect of these differences is that for a given $r_e$ and



SWE, snow broadband albedo is larger for the OHC than for spheres (Fig. 1c). The difference is particularly pronounced for
thin snowpacks with a large snow grain size (up to 0.125 for SWE=$10\,\mathrm{kg\,m^{-2}}$ and $r_e$=$1500\,\mathrm{\mu m}$).

## 3 Experiments

Results from a total of six experiments are discussed in this paper (Table 1). Most of the discussion is focused on the experiments SPH and NONSPH. The only difference between these experiments is that SPH employs the spherical snow grain shape assumption and NONSPH the OHC shape assumption. To support the interpretation of the climatic differences between NONSPH and SPH, two experiments were conducted in which the OHC shape assumption was applied only over land (NON-SPHLND) or only over ice (NONSPHICE). In the fifth experiment, called TUNED, the OHC shape assumption is applied, but the parameterization of snow grain size is modified, in an attempt to minimize the climatic differences to the SPH experiment. This experiment will be detailed in Sect. 5. The sixth experiment SPH2XCO2 was conducted for evaluating the equilibrium climate sensitivity of the current version of NorESM. This experiment was otherwise identical to SPH but used a doubled atmospheric $CO_2$ concentration ($734\,\mathrm{ppmv}$) instead of the year 2000 value $367\,\mathrm{ppmv}$ used in the other experiments. The concentrations of other greenhouse gases and aerosol emissions (Iversen et al., 2013; Kirkevåg et al., 2013) were representative of year 2000 in all experiments.

The length of the experiments was 100 years except for TUNED, which was branched off from SPH after 30 simulated years and then run for 70 more years. For each experiment, the last 60 years were used for the analysis of the model climatology. To evaluate the statistical significance of the differences between the experiments, a two-sided $t$ test was used. Autocorrelation between subsequent years was accounted for using the method of Zwiers and von Storch (1995).

## 4 Results

### 4.1 Instantaneous radiative effect of changed snow grain shape

The change of snow grain shape in the model exerts an instantaneous change in shortwave (SW) radiative fluxes. This flux change results from a change in model parameterization rather than from a physical perturbation that could actually occur in nature, but from the point of view of interpreting the model results, we can think of it as a radiative forcing (RF) to which the simulated climate responds. The RF was quantified through diagnostic radiation calculations, by calling the radiation scheme twice at each radiative time step (once for the spherical shape assumption and once for the OHC shape assumption). Figure 2 displays the annual-mean RF of the net (down−up) SW flux at the top of the model (TOM) atmosphere diagnosed in the SPH experiment.

Since snow albedo is larger for the OHC than for spheres (Fig. 1c), the RF is negative (or zero, in permanently snow-free regions). However, its spatial distribution is highly inhomogeneous. The largest negative values occur where high insolation coincides with abundant snow cover. Thus, the RF reaches $-4\,\mathrm{W\,m^{-2}}$ in parts of Tibet and Antarctica and $-2$ to $-3\,\mathrm{W\,m^{-2}}$ over Greenland. Over large parts of the Arctic Ocean, in the Southern Ocean close to Antarctica, and in the northernmost parts





of Siberia the RF ranges between $-1$ and $-2\,\mathrm{W\,m^{-2}}$. In comparison, the RF is much smaller, for example, in the southern

parts of northern Eurasia (at $\sim 60°$N), mainly because the change in snow albedo is largely masked by forests. In the northern

hemisphere, the largest negative RF values generally occur in spring just before snow melt, reaching locally $-10\,\mathrm{W\,m^{-2}}$ in

Tibet in April. Over the permanently snow-covered Antarctica, the RF peaks in December, with a local maximum negative

value of $-13\,\mathrm{W\,m^{-2}}$. The global annual-mean RF is $-0.218\,\mathrm{W\,m^{-2}}$, with estimated contributions of $-0.144\,\mathrm{W\,m^{-2}}$ from

snow over land and $-0.074\,\mathrm{W\,m^{-2}}$ from snow over sea ice. The RF defined in terms of net SW flux at the surface is slightly

larger than that at the TOM, with a global annual-mean of $-0.233\,\mathrm{W\,m^{-2}}$, but the spatial distribution is nearly identical.

In principle, the choice of the SPH experiment as the baseline for evaluating the RF is arbitrary. Alternatively, if one selects

the NONSPH experiment as the baseline, a slightly larger estimate of the RF is obtained, due to larger snow and sea ice cover

in NONSPH than in SPH (see Sect. 4.3). In that case, the global-mean TOM RF becomes $-0.276\,\mathrm{W\,m^{-2}}$, with contributions

of $-0.164\,\mathrm{W\,m^{-2}}$ from snow over land and $-0.112\,\mathrm{W\,m^{-2}}$ from snow over sea ice.

## 4.2 Temperature and precipitation

Although the global-mean RF associated with the changed snow grain shape is modest, it has a substantial impact on the

climate simulated by NorESM. The global annual-mean 2-m air temperature ($T_2$) in NONSPH is $1.17\,\mathrm{K}$ lower than in SPH

(Fig. 3a). The largest differences occur at high latitudes, reaching $-4\,\mathrm{K}$ in the extreme northeastern parts of Russia and as

much as $-7\,\mathrm{K}$ in the Southern Ocean off the coast of Antarctica at the $0°$ E longitude. As expected, when the snow grain shape

is changed only over land (experiment NONSPHLND) the cooling compared to SPH is generally somewhat larger over land

than over ocean (Fig. 3c), and vice versa for the experiment NONSPHICE, in which the snow grain shape is changed only

over sea ice (Fig. 3e). The global-mean temperature difference to SPH is just slightly larger for NONSPHLND ($-0.58\,\mathrm{K}$) than

for NONSPHICE ($-0.54\,\mathrm{K}$), and the sum of these contributions ($-1.12\,\mathrm{K}$) is slightly less negative than the actual difference

between NONSPH and SPH ($-1.17\,\mathrm{K}$). Recalling from Sect. 4.1 that most of the RF associated with changed snow grain shape

comes from land areas (59% or 66%, depending on the choice of the baseline), these numbers imply that in a relative sense,

the present-day climate simulated by NorESM is more sensitive to changing the snow grain shape over sea ice than over land.

Figure 4 compares the annual and seasonal-mean $T_2$ in the SPH and NONSPH experiments with ERA-Interim reanalysis

data (Dee et al., 2011) for years 1990–2014. This comparison must be taken with some caution, especially because the $Q$-

fluxes employed in the slab ocean model are based on a preindustrial simulation. The slab ocean configuration of the model

is thus targeted at approximating the preindustrial equilibrium climate of NorESM when run with preindustrial atmospheric

composition. Correspondingly, in our experiments with near-present atmospheric composition, the simulated climate represents

an equilibrium consistent with near-present forcings. However, the observed climate is in a state of transient warming, which

is colder than the equilibrium climate. Therefore, the present simulations would be (ideally) expected to feature a slight warm

bias compared to observations.

In the SPH experiment, the global annual-mean $T_2$ ($288.05\,\mathrm{K}$) is $0.46\,\mathrm{K}$ higher than in ERA-Interim, with most notable

positive biases of 2–4 K over the Southern Ocean, roughly between $35°$ S and $60°$ S (Fig. 4a). Referring to the above discussion,

these features are probably at least partly caused by the experimental setup. In contrast, the global annual-mean $T_2$ in NONSPH



(286.88 K) is 0.71 K lower than in ERA-Interim. While a slight warm bias prevails in the northern parts of the Southern Ocean,
substantial annual-mean cold biases of up to 4–6 K occur in the Arctic Ocean, in the northernmost parts of North America
and Russia, and in the southernmost parts of the Southern Ocean (Fig. 4b). Seasonally, these biases are most severe in autumn
(March–May in the southern hemisphere (SH) and September–November in the northern hemisphere (NH)), reaching $-10$ to
$-8$ K in some areas (Fig. 4h,j). This is related to expanded sea ice cover and an increased frequency of surface inversions in the
NONSPH experiment. Furthermore, NONSPH features a strong cold bias compared to ERA-Interim in the central Antarctica
in summer, which is when the radiative effects of changed snow grain shape are largest (Fig. 4g). The differences between
NONSPH and ERA-Interim locally reach $-11$ K, while the differences between NONSPH and SPH reach $-7$ K.

Apart from $T_2$, there are notable differences between the NONSPH and SPH experiments in many other climate parameters.
As an example, the differences in precipitation are shown in Fig. 3b. The global-mean precipitation is smaller in NONSPH than
in SPH by $0.062\,\mathrm{mm\,d^{-1}}$, or 2.1%. In particular, there is a widespread reduction in precipitation at middle and high latitudes.
This is consistent (though opposite in sign) with the increased global and mid-to-high latitude precipitation seen in response
to global warming in experiments forced with increasing greenhouse gas concentrations (Collins et al., 2013). Interestingly,
a band of negative precipitation differences appears just south of the equator, along with positive differences on the northern
side, indicating a northward shift of the ITCZ in the NONSPH experiment. The northward shift of the ITCZ is also found in
NONSPHICE (Fig. 3f) but much less so in NONSPHLND (Fig. 3d). ITCZ response to extratropical thermal forcing has been
discussed in several previous studies (e.g. Chiang and Bitz, 2005; Kang et al., 2008, 2009; Kirkevåg et al., 2008; Frierson
and Hwang, 2012; Iversen et al., 2013). Generally, the ITCZ moves towards the hemisphere experiencing relative warming,
which is also the case here, since NONSPH experiences more cooling compared to SPH in the SH ($-1.34$ K) than in the NH
($-1.00$ K). Furthermore, the hemisphere-mean difference in TOM net radiation between NONSPH and SPH is negative in
the SH (ca. $-0.2\,\mathrm{W\,m^{-2}}$) and positive in the NH (ca. $0.2\,\mathrm{W\,m^{-2}}$), which requires an ITCZ shift towards the NH, in order
to induce an anomalous cross-equatorial atmospheric heat transport towards the SH (Kang et al., 2008, 2009). It should be
noted, however, that the use of a slab ocean model precludes changes in cross-equatorial ocean heat transport, which may act
to exaggerate the ITCZ shift (Kay et al., 2016).

### 4.3   Snow, sea ice and surface albedo

The large sensitivity of NorESM to changed snow grain shape is related to strong snow-albedo and sea-ice-albedo feedbacks,
which greatly amplify the change in surface albedo. To demonstrate this point, Fig. 5a shows the albedo difference between
OHC and spheres from diagnostic radiation calulations in the SPH experiment, and Fig. 5b the actual ("interactive") albedo
difference between the NONSPH and SPH experiments. In each case, the albedo is calculated as the ratio of annual-mean
upwelling and downwelling SW fluxes at the surface. In the diagnostic case, the largest albedo differences (0.02–0.03) occur
over permanently snow-covered regions (Antarctica, Greenland, and central parts of the Arctic Ocean). In the interactive case,
where the OHC shape assumption impacts the model integration, the albedo differences are much larger, locally 0.1–0.2 in the
northernmost parts of North America and Eurasia and in the adjacent Arctic Ocean, and up to 0.2–0.4 in parts of the Southern
Ocean. The difference in global-mean energy-weighted surface albedo in the interactive case is 0.0113 (0.1329 for NONSPH



vs. 0.1216 for SPH), as compared with 0.0017 in the diagnostic case. This indicates that feedback effects in NONSPH amplify the instantaneous albedo difference between the OHC and the spherical shape assumptions by a factor of more than 6.

The large albedo differences between NONSPH and SPH are mainly related to increased snow and sea ice cover in the NONSPH experiment (Fig. 6). The difference in annual-mean snow fraction between NONSPH and SPH exceeds 0.05 in many regions in the northern hemisphere, including Alaska and parts of western USA, Scandinavia, northern Russia and Tibet. In broad terms, this means that the season with snow cover is extended by 18 days or more. Much larger differences of up to 0.15–0.3 are found in the Canadian Arctic Archipelago, in the Taymyr region and in the northeastern tip of Russia (the

Chukotka Peninsula). In fact, in these Arctic regions, the ground is (unrealistically) snow-covered throughout the year in the NONSPH experiment. Sea ice cover is also extended in the NONSPH experiment, most strikingly in the Southern Ocean. The maximum difference in ice cover (at 63° S, 0° E) is as large as 0.67 (0.85 in NONSPH vs. 0.18 in SPH) and is associated with maximum differences in both surface albedo (Fig. 5b) and 2-m air temperature (Fig. 3a).

In addition to snow and sea ice area, a smaller contribution to the albedo differences between NONSPH and SPH arises

from changes in snow microphysics. For reference, Fig. 7a displays the average value of the top-layer (i.e., the uppermost 2 cm) snow grain effective radius $r_e$ simulated by SNICAR in the SPH experiment. The values vary widely, from 70–100 μm in central Antarctica and Greenland (where snow metamorphism is slow due to the prevailing cold conditions) to over 350 μm (e.g.) in southern Russia. The corresponding relative differences between NONSPH and SPH in Fig. 7b vary in sign depending on location. In general, it is not trivial to interpret these differences since they are influenced by several factors (i.e., changes

in temperature, precipitation, and the length of the snow season). However, in Antarctica and in much of the Arctic, $r_e$ is consistently smaller in NONSPH than in SPH. This stems from the lower temperatures and especially the rarer occurrence of melting conditions in NONSPH, which reduce the rate of snow metamorphism as compared with SPH. All other factors being equal, this leads to reduced snow grain size and increased snow albedo. The impact of this is best seen over the permanently snow-covered Antarctica, where the region with albedo differences larger than 0.03 is substantially larger in Fig. 5b than in

Fig. 5a.

### 4.3.1 Comparison with observations

Here, the snow cover simulated by NorESM is compared with the National Oceanic and Atmospheric Administration (NOAA) NH snow cover extent (SCE) climate data record (CDR) (Robinson et al., 2012; Estilow et al., 2015) and the sea ice cover with the NOAA National Snow and Ice Data Center (NSIDC) passive microwave sea ice concentration CDR (Meier et al.,

2013; Peng et al., 2013). Figure 8 displays the spatial distribution of the differences between each of the SPH and NONSPH experiments and these observations for the months of February and August, which correspond to near-maximum and near-minimum snow and ice coverage in the NH, and vice versa in the SH. The seasonal cycles of NH snow cover area and both NH and SH sea ice cover are displayed in Figure 9. The main findings are as follows.

First, regarding land snow cover in winter, the difference between NONSPH and SPH is relatively small. Both show similar

differences from the NOAA SCE data in February: underestimation in parts of North America and western Eurasia, but over-estimation in Tibet (Fig. 8a–b). The net effect is that the NH winter snow cover simulated by NorESM is slightly smaller than





that in the NOAA SCE data (Fig. 9a), with a larger underestimation for SPH. This result is, however, sensitive to the dataset used. A comparison of the NorESM results with the European Space Agency's GlobSnow dataset (Metsämäki et al., 2015) suggested that NorESM overestimates rather than underestimates the NH winter snow cover, although this dataset does not
cover regions/months with very low solar elevations.

Second, regarding sea ice in winter, the SPH experiment underestimates rather substantially the sea ice cover in much of the Southern Ocean (Fig. 8c) and therefore also the sea ice area integrated over the SH (Fig. 9c). This is consistent with the positive temperature bias seen in Fig. 3e (which might be partly an artifact of the experimental setup, as discussed in Sect. 4.2). In the NONSPH experiment, however, the SH winter ice concentration is overestimated slightly. The NH sea ice cover in winter is
slightly higher in NorESM than in the NSIDC ice concentration data, more so for NONSPH than for SPH (Figs. 8a–b and 9b).

Third, in contrast to the somewhat mixed winter results, it is seen from Figs. 8 and Fig. 9 that in summer, SPH agrees much better with the observations than NONSPH does. The SPH experiment reproduces quite closely the observed summertime minima of NH land snow cover and both NH and SH sea ice cover (Fig. 9a–c). The excellent agreement could be partly fortuitous, as the simulations represent, in principle, the climatic equilibrium for a near-present atmospheric composition
rather than the observed transient climate (cf. Sect. 4.2). At any rate, the NONSPH experiment overestimates these minima substantially: by $2.3 \times 10^6 \, \mathrm{km}^2$ (or ca. 90%) for the NH land snow cover, by $2.9 \times 10^6 \, \mathrm{km}^2$ (ca. 60%) for the NH sea ice cover, and most strikingly, by $6.5 \times 10^6 \, \mathrm{km}^2$ (ca. 310%) for the SH sea ice cover. The overestimation of NH land snow cover in NONSPH originates from the Canadian Arctic Archipelago, the Taymyr peninsula and the northeastern tip or Russia, which unrealistically feature permanent snow cover in NONSPH (Fig. 8d).

Having noted the differences in snow and sea ice cover, we turn our attention to an even more fundamental question: how do the simulated snow albedos in NONSPH and SPH compare with observations? To address this question, we consider the surface albedo over Antarctica and Greenland, focusing on inland regions with permanent snow cover and no vegetation (Fig. 10). Annual-mean clear-sky broadband shortwave albedos in the SPH and NONSPH experiments are compared with three datasets based on satellite observations: MODIS MCD43C3.005 (Schaaf et al., 2011) black-sky albedos (keeping only the best
quality data with a quality flag of 0), CERES EBAF-Surface Ed2.8 (Kato et al., 2013) clear-sky albedos, and CLARA-A2 (Karlsson et al., 2017) black-sky albedos, where "black-sky albedo" refers to the albedo for direct solar illumination in the absence of an atmosphere. These albedo products were averaged from their generic spatial resolution to the NorESM grid and over 10 years in time (March 2000 – Feb 2010 for MODIS and CERES, and Jan 2000 – Dec 2009 for CLARA-A2). For reference, area-mean albedos averaged over two regions denoted as Central Greenland and Central Antarctica (marked with
the black contours in Fig. 10) are given in the panel titles of Fig. 10.

In agreement with Fig. 5, the NONSPH experiment features systematically higher snow albedo than SPH, typically by approximately 0.03 (Fig. 10a,b,f,g). Furthermore, over Greenland, all three satellite datasets agree quite closely (Fig. 10c–e) while over Antarctica, the CERES EBAF values are substantially lower than those in MODIS and CLARA-A2, on average by ca. 0.04 (Fig. 10h–j). Strikingly, the albedo values in the NONSPH experiment exceed those in all three satellite datasets,
typically by 0.04–0.05. In fact, even in the SPH experiment, the area-mean surface albedos are slightly higher than in the satellite data, although over Antarctica, the differences to MODIS and CLARA-A2 are rather marginal.





The comparison with the satellite albedo datasets strongly suggests that the NONSPH experiment overestimates snow albedo, at least over Antarctica and Greenland. Further support for this hypothesis is obtained by comparing the modelled reflected solar radiation at the top of the atmosphere ($F_{\mathrm{TOA}}^{\uparrow}$) with CERES EBAF-TOA Ed2.8 data (Loeb et al., 2009) (Table 2). In NONSPH, $F_{\mathrm{TOA}}^{\uparrow}$ is larger than in CERES EBAF by over 8 W m$^{-2}$ over Central Antarctica and by over 3 W m$^{-2}$ over Central Greenland, irrespective of whether all-sky or clear-sky fluxes are considered. The corresponding differences from CERES EBAF are substantially smaller for SPH. These differences are influenced by atmospheric absorption and scattering, and there are uncertainties both in the simulated cloud fields and in satellite cloud detection over snow surfaces (which could affect the CERES EBAF clear-sky fluxes). Even so, in principle, overestimated $F_{\mathrm{TOA}}^{\uparrow}$ is consistent with the suggestion that snow albedo is overestimated in NONSPH.

## 5   A tuning exercise

Above, it has been shown that changing the snow grain shape assumption in NorESM has a substantial impact on the simulated climate. In general, the use of the OHC (i.e., non-spherical) shape assumption results in larger climatic biases than the use of the default spherical shape assumption, including a substantial cold bias at high latitudes (Fig. 4) and overestimated land snow cover and sea ice cover in the warm season (Figs. 8 and 9). While somewhat disappointing, this is not particularly surprising. Very frequently, even physically motivated parameterization changes lead to some deterioration of the simulate climate, if not accompanied by model retuning (e.g. Hourdin et al., 2017).

In this section, we explore one option for retuning: the reduction of snow albedo through adjusting the snow grain effective size $r_e$. This is motivated by Fig. 10 and Table 2, which indicate that the NONSPH experiment probably overestimates snow albedo (at least in Greenland and Antarctica), and also by the fact that the snow grain $r_e$ is a relatively poorly known parameter. Here, we set a simple target for model retuning: we aim at reproducing the albedo simulated in the SPH experiment when using the OHC shape assumption for snow grains. Since snow albedo decreases with increasing snow grain size (Fig. 1c), this can be achieved by using a larger $r_e$ in connection with the OHC shape assumption. Based on diagnostic radiation calculations conducted for a three-year period of the SPH experiment, it was found that the global-mean difference between the OHC and spherical shape assumptions in both the surface and TOM net solar radiation is minimized, if for the OHC, the values of $r_e$ are multiplied by approximately 1.7.

Based on the above considerations, a retuned model version using the OHC was constructed with values of $r_e$ increased by ca. 70%. To achieve this, the limiting values of snow grain $r_e$ were increased by 70% from their default values: to $r_{e,\mathrm{nonmelt}} = 850\,\mu\mathrm{m}$ and $r_{e,\mathrm{melt}} = 2550\,\mu\mathrm{m}$ for snow on sea ice, and to $r_{e,0} = 92.7\,\mu\mathrm{m}$, $r_{e,\mathrm{rfrz}} = 1700\,\mu\mathrm{m}$ and $r_{e,\mathrm{rmax}} = 2550\,\mu\mathrm{m}$ for snow on land (the physical meaning of these parameters is explained in Sect. 2). Furthermore, to get a roughly 70% increase in the actual values of $r_e$ on land, the snow grain growth rate $dr_e/dt$ also had to be increased on average by $\approx$70%. For this end, the snow grain growth rate tuning constant was set to xdrdt = 2.3. Since $dr_e/dt$ depends non-linearly on $r_e$ (Eqs. 2 and 3), this value was determined by trial and error (i.e., by analyzing the values of $r_e$ in additional short NorESM experiments).





Figure 11 compares the results of the TUNED experiment (with the modifications noted above) with the SPH experiment.

First, Fig. 11a confirms that in most land regions, the relative difference between the TUNED and SPH experiments in the time-mean snow grain $r_e$ in the uppermost snow layer is close to 70% (the difference in the snow-fraction weighted area-mean value being 69.7%). Second, Fig. 11b shows that the differences in 2-m air temperature between TUNED and NONSPH are generally small. The mean and rms differences ($-0.01$ K and $0.13$ K) are much smaller than those between the NONSPH and SPH experiments ($-1.17$ K and $1.62$ K; see also Fig. 3a). The differences in snow and sea ice cover are also minor (Fig. 11c,d).

Both the NH snow cover area and the NH and SH sea ice area are just marginally larger in TUNED than in SPH (Fig. 9).

The tuning approach applied here is not necessarily the optimal one when using the OHC shape assumption in climate simulations. In particular, it results in values of snow grain $r_e$ over sea ice that might be unrealistically large. In fact, the default value of $r_{e,\text{nonmelt}} = 500\,\mu\text{m}$ in NorESM equals the upper limit of this parameter considered by Urrego-Blanco et al. (2016), suggesting that the use of $r_{e,\text{nonmelt}} = 850\,\mu\text{m}$ in the TUNED experiment is not really justified. Even so, the TUNED

experiment serves to demonstrate an important point: for climate simulations, the effects of changing the snow grain shape assumption are almost indistuingishable from the effects of changing the snow grain size.

## 6  Radiative effect of absorbing aerosols in snow

The SNICAR scheme simulates the concentration of absorbing aerosols in snow (hydrophobic and hydrophilic black carbon (BC) and mineral dust in four size bins) based on aerosol deposition fields produced by the NorESM aerosol scheme. The

grid-mean surface radiative effect (RE) due to absorbing aerosols in snow diagnosed by SNICAR is shown in Fig. 12a for the SPH experiment. The distribution of the aerosol RE is highly inhomogeneous with maxima of ca. $6\,\text{W}\,\text{m}^{-2}$ in Tibet (due to both BC and dust), ca. $4\,\text{W}\,\text{m}^{-2}$ in eastern Mongolia / northeastern China (mainly due to dust), and ca. $2\,\text{W}\,\text{m}^{-2}$ in eastern Greenland (mainly due to BC). The global land-area mean RE is $0.191\,\text{W}\,\text{m}^{-2}$, corresponding to $0.056\,\text{W}\,\text{m}^{-2}$ averaged over the entire planet, with roughly equal contributions from BC and mineral dust.

Before discussing the differences between the NONSPH and SPH experiments, it is instructive to first consider selected off-line radiative transfer results for the effects of absorbing aerosols on the albedo of snow, as shown in Fig. 13 (see the figure caption for details). For optically thick snow (the case with SWE=$100\,\text{kg}\,\text{m}^{-2}$), the albedo reduction due to aerosols increases with increasing effective snow grain size $r_e$ and is larger for spheres than for the OHC, irrespective of the aerosol type considered. Both these features are consistent with the calculations of Dang et al. (2016, their Fig. 6) for spherical and

non-spherical snow grains. While the transfer of solar radiation in snow is also influenced by absorption, these features can be understood qualitatively by considering the transport path length (Kokhanovsky and Zege, 2004),

$$l_{\text{tr}} = \frac{1}{\sigma_{\text{ext}}(1-g)}. \tag{5}$$

The extinction coefficient $\sigma_{\text{ext}}$ is inversely proportional to the snow grain $r_e$. Therefore, the larger $r_e$ is, the deeper the radiation penetrates, which results in increased chances of aerosol absorption. Furthermore, as shown in Fig. 1a, $g$ is considerably smaller

for the OHC than for spheres, especially in the visible spectral band, which dominates the absorption by BC and mineral dust





in snow ($g \approx$ 0.77–0.78 for the OHC vs. $g \approx$ 0.89 for spheres). Therefore, for snow grains with the same $r_e$, radiation penetrates deeper into snow for the spherical shape assumption than for the OHC shape assumption, which results in more aerosol absorption and a larger albedo reduction for spheres. It is further noted from Fig. 13 that these arguments only apply to the case of a relatively thick snowpack. For a thin snowpack (the case with SWE=$10\,\mathrm{kg\,m^{-2}}$), the albedo reduction depends little on the snow grain size and shape for BC aerosols (Fig. 13a) while for dust (Fig. 13b), the albedo reduction in fact decreases slightly with increasing snow grain size, especially for spherical snow grains. This is related to scattering (rather than absorption) by the dust particles.

Overall, Fig. 13 suggests that the use of the OHC shape assumption in the NONSPH experiment should lead to a reduced RE by absorbing aerosols in snow, as compared with the SPH experiment. Figure 12b confirms that the aerosol RE is indeed smaller for NONSPH than for SPH over Greenland and Antarctica, which are covered by thick snow throughout the year. Interestingly, however, the opposite is true in much of North America and Eurasia: the RE is larger in the NONSPH experiment than in the SPH experiment. Even the global-mean RE is slightly larger in NONSPH than in SPH (by 14%). This primarily results from the longer snow season in the NONSPH experiment (Fig. 6), which exposes the snowpack to more incident solar radiation especially in spring, and, in the Canadian Arctic Archipelago, the Taymyr peninsula and the northeastern tip or Russia, throughout the summer. Thus, changes in snow conditions can swamp the direct effect of changed snow grain shape on aerosol absorption in snow.

Furthermore, the blue curves in Fig. 13 represent the case in which the snow grain $r_e$ is 70% larger for the OHC than for spheres, thus approximating the tuning of snow grain size in the TUNED experiment. Even in this case, the albedo reduction due to absorbing aerosols is slightly smaller for the OHC than for spheres in the case of a thick snowpack (SWE=$100\,\mathrm{kg\,m^{-2}}$). Consistent with this, the TUNED experiment features slightly smaller aerosol RE than the SPH experiment in most of Greenland and Antarctica, although the difference between TUNED and SPH (Fig. 12c) is smaller than that between NONSPH and SPH (Fig. 12b). Slight negative differences also dominate in the northernmost parts of North America and Eurasia, while in the central parts, some positive differences are simulated. The resulting global land-area mean RE due to absorbing aerosols in snow in the TUNED experiment ($0.187\,\mathrm{W\,m^{-2}}$) is just slightly below that in SPH ($0.191\,\mathrm{W\,m^{-2}}$).

# 7 Discussion

In this section, some of the primary findings of this work are discussed in the light of the previous literature.

## 7.1 High efficacy of the radiative forcing associated with changed snow grain shape

It was found in Sect. 4.2 that the climate simulated by NorESM is quite sensitive to changed snow grain shape, especially in relation to the rather modest global-mean instantaneous "radiative forcing" associated with this change (Sect. 4.1). This can be quantified using the concept of efficacy (Hansen et al., 2005), defined as

$$E_i = \lambda_\mathrm{i}/\lambda_\mathrm{CO2}, \tag{6}$$





where $\lambda_i$ and $\lambda_{\mathrm{CO2}}$ are sensitivity parameters (ratio of global-mean $T_2$ change to radiative forcing RF) for a forcing mechanism $i$ and for changed $CO_2$ concentration, respectively:

$$\lambda_i = \Delta T_{2,i}/RF_i, \tag{7}$$

$$\lambda_{\mathrm{CO2}} = \Delta T_{2,\mathrm{CO2}}/RF_{\mathrm{CO2}}. \tag{8}$$

In the case of changed snow grain shape, $\lambda_{\mathrm{grainshape}} = 1.17\,\mathrm{K}/0.218\,\mathrm{W\,m^{-2}} = 5.36\,\mathrm{K}\,\left(\mathrm{W\,m^{-2}}\right)^{-1}$ (or alternatively, if one uses the NONSPH experiment as the baseline to define the RF, $\lambda_{\mathrm{grainshape}} = 1.17\,\mathrm{K}/0.276\,\mathrm{W\,m^{-2}} = 4.22\,\mathrm{K}\,\left(\mathrm{W\,m^{-2}}\right)^{-1}$). For comparison, the global-mean temperature difference between the SPH2XCO2 and SPH experiments (resulting from a doubling of the atmospheric $CO_2$ concentration from 367 to 734 ppmv) is 3.47 K. The corresponding RF was evaluated using the fixed-SST method (Eq. (1) in Hansen et al. (2005)) as $4.05\,\mathrm{W\,m^{-2}}$. This yields $\lambda_{\mathrm{CO2}} = 3.47\,\mathrm{K}/4.05\,\mathrm{W\,m^{-2}} = 0.86\,\mathrm{K}\,\left(\mathrm{W\,m^{-2}}\right)^{-1}$. Therefore the efficacy of the RF associated with snow grain shape is as high as 4.9–6.2. This is broadly consistent with, but even higher than, the efficacy of RF due to BC in snow reported by Flanner et al. (2007), $E \approx 2.1$–4.5. Flanner et al. (2007) attribute the high efficacy of BC in snow to the fact that the forcing generally peaks in the local spring-time, coincident with the melt onset, when it is able to trigger rapid snow ablation and snow-albedo feedbacks. These feedback processes are certainly also at play in the current NorESM experiments, but it is not fully clear why the efficacy is even higher in our experiments than in the experiments of Flanner et al. (2007).

While there are differences in the model and experimental setup between this study and Flanner et al. (2007), perhaps the most obvious difference lies in the spatial distribution of RF. The RF due to BC in snow considered by Flanner et al. (2007) is mainly confined to the mid-to-high latitudes of the NH, but the RF due to changed snow grain shape (Fig. 2) is roughly equally split between the two hemispheres, with contributions of 51% from the SH and 49% from the NH. Furthermore, the negative temperature difference between the NONSPH and SPH experiment is larger in the SH ($-1.34\,\mathrm{K}$) than in the NH ($-1.00\,\mathrm{K}$), most probably due to the large sea ice response in the Southern Ocean (Figs. 6 and 8). It is also worth noting that the temperature response due to changed snow grain shape is larger at the surface level than in the free troposhere (i.e., it is enhanced by a positive lapse-rate feedback). This is especially true at polar latitudes, but also the global-mean temperature difference between NONSPH and SPH is larger at the 2-m level ($-1.17\,\mathrm{K}$) than in the mid-troposphere (ca. $-0.8\,\mathrm{K}$). However, in lack of information on vertical temperature response in Flanner et al. (2007), it cannot be verified whether a more positive lapse-rate feedback contributes to the higher efficacy found in the present work.

### 7.2 Scaling factor between the size of non-spherical and spherical snow grains

It was shown in Sect. 5, both through diagnostic radiation calculations and climate simulations, that the effects of replacing the spherical snow grain shape assumption with the non-spherical OHC assumption can be largely offset by increasing the snow grain effective radius $r_e$. The use of the OHC assumption with $r_e$ multiplied by roughly 1.7 led to results almost indistuingishable from those obtained with the spherical shape assumption.

Recently, Dang et al. (2016) demonstrated that the albedo of a snowpack consisting of non-spherical snow grains can be mimicked by using smaller grains of spherical shape. They derived the scaling factor between the size of non-spherical and





spherical snow grains with the same albedo by using, in the non-spherical case, the parameterization of Fu (2007) for the asymmetry factor, which assumes hexagonal ice crystals with rough surfaces. The derived scaling factor depends mainly on the aspect ratio (AR) of the non-spherical grains, varying between approximately 1.2 and 2.4. The largest values occur for equidimensional snow grains (AR≈1) (Dang et al., 2016, their Fig. 9), because $g$ is smallest for these (for example, at $\lambda = 0.5\,\mu m$, $g \approx 0.74$, as compared with $g \approx 0.89$ for spheres).

While the present estimate of 1.7 falls well in the range of scaling factors derived by Dang et al. (2016), it is worth noting that their analysis might actually overestimate the scaling factors somewhat. Dang et al. (2016) only accounted for the difference in asymmetry parameter between the non-spherical and spherical case, assuming the same extinction coefficient and single-scattering albedo in both cases. However, both single-scattering modelling (e.g., Kokhanovsky and Zege, 2004; Räisänen et al., 2015) and observational analysis for real snowpacks (Libois et al., 2014) suggest that the co-albedo $\beta$ (or the absorption

enhancement parameter $B$, which is directly proportional to $\beta$), is typically somewhat larger for non-spherical than spherical snow grains. As noted in Sect. 2.1, for the present treatment of non-sphericity based on the OHC, $\beta$ exceeds that for spheres by 12–30%, depending on $r_e$ and spectral band, except for the most strongly absorbing band 1.5–5.0 $\mu m$ (see Fig. 1b). The impacts of this can be appreciated by considering the expression for the spectral spherical albedo of an optically thick (i.e., semi-infinite) weakly absorbing snowpack (Kokhanovsky, 2013):

$$A = \exp\left(-4\sqrt{\frac{\beta}{3(1-g)}}\right). \tag{9}$$

Therefore, while the smaller $g$ of non-spherical snow grains leads to a snow albedo higher than that for spheres, their larger $\beta$ partially compensates for this difference. In fact, within the limits of validity of Eq. (9), fractional differences in $\beta$ and $1-g$ are equally important for snow albedo and the scaling factor for snow grain effective radius.

### 7.3   Uncertainties in snow albedo calculation

The analysis conducted for Greenland and Antarctica in Sect. 4.3.1 suggests that the NONSPH experiment overestimates the surface albedo of purely snow-covered regions. Several uncertainty factors can be identified in the calculation of snow albedo.

- A key uncertainty lies in the size-dependent snow grain single-scattering properties, especially the asymmetry parameter $g$. For the OHC employed in the current work, $g \approx 0.78$ at weakly absorbing wavelengths. This value is based on direct measurements of angular scattering for two cases of blowing snow, that is, snow grains detached by wind from
the snow surface. It is not clear how well these measurements represent $g$ for grains deeper in the snowpack. To our knowledge, no direct measurements of $g$ exist for snow staying on ground. However, using an indirect technique based on multidirectional polarization measurements (van Diedenhoven et al., 2012), Ottaviani et al. (2015) derived values of $g = 0.84$, 0.876 and 0.90 at $\lambda = 864\,\mu m$ from aircraft measurements over three snowfields in Colorado. These values are, in fact, closer to the $g$ of large spherical snow grains ($g \approx 0.89$) than that of the OHC. Therefore, should the values
derived by Ottaviani et al. (2015) be typical of snow, the use of the OHC would tend to overestimate snow albedo.





- Another important uncertainty factor is the snow grain effective radius ($r_e$). In the absence of a well-established observational climatology of $r_e$, we have not attempted to validate the simulated $r_e$ in the present work. In principle, a high bias in simulated snow albedo could result from a low bias in $r_e$.

- Snow albedo is reduced by absorbing aerosols in snow. The analysis by Jiao et al. (2014) suggests that CAM4-Oslo (the atmospheric component of NorESM) tends to underestimate slightly (by ∼20%) the BC concentration in snow in the Arctic. However, the albedo reduction depends not only on BC but also on dust, on the seasonal cycles of the aerosol concentrations, the aerosol optical properties and their mixing state in snow, and the physical properties of the snowpack (cf. Fig. 13). Further analysis of this complex topic is left for future study. In the present context, it is worth noting that at least in the NorESM simulations, the effect of aerosols on snow albedo is generally smaller than the effect of snow grain shape (the albedo difference between the OHC and spherical shape assumptions), on average by a factor of three. For example, in the Central Greenland region in Fig. 10, the average albedo reduction due to aerosols is 0.009 for SPH and 0.005 for NONSPH, while in the near-pristine Central Antarctica, it is well below 0.001.

- While clear-sky albedos were considered in Fig. 10 for better compatibility with satellite measurements, it is worth noting that snow albedo depends on cloudiness. Generally, clouds act to increase the snow broadband albedo, by depleting the incoming solar radiation most efficiently at near-infrared wavelengths where the snow albedo is relatively low. The magnitude of this effect appears to be broadly similar in NorESM and in the CERES EBAF dataset. For example, the average difference between all-sky and clear-sky surface albedo in the Central Greenland region of Fig. 10 is 0.026, 0.024 and 0.022 for the SPH experiment, the NONSPH experiment and the CERES EBAF dataset, respectively, while the corresponding differences for the Central Antarctica region are 0.010, 0.008, and 0.016.

- A small positive albedo bias may result from the two-stream approximation employed in SNICAR. Compared to reference computations made with DISORT (Stamnes et al., 1988) with 32 streams, the SNICAR approach overestimates the broadband albedo of thick snow for isotropic (i.e., completely diffuse) incident radiation typically by ∼0.008, the positive bias coming from the near-IR bands where the delta-hemispheric mean approximation (Toon et al., 1989) is used. The errors for direct solar radiation depend on the solar elevation but are generally smaller, so the bias in all-sky and especially clear-sky snow albedo is probably smaller than that for isotropic incident radiation.

- The albedo calculation in SNICAR assumes a plane-parallel horizontally homogeneous medium, which means that snow surface structure is ignored. Surface characteristics such as sastrugi have been shown to slightly reduce the snow albedo at near-infrared wavelengths (Warren et al., 1998), by a few percent at low solar elevations.

- Finally, an underlying assumption in the radiative transfer calculations is that light scattering and absorption of snow grains is independent of surrounding grains. In reality, snow is a close-packed medium. He et al. (2017) report, based on calculations for clusters of spherical snow grains, that close-packing can reduce the albedo of pure snow by ca. 0.01 in the visible region and by up to 0.05 in the near-infrared, with even larger effects for dirty snow. This reduction appears to come from an increase of effective snow grain size (i.e., the ratio of volume to projected area), which decreases the snow




extinction coefficient and increases the co-albedo. The asymmetry parameter is actually reduced by the close-packed effects (as also noted by Kokhanovsky, 1998), which, if acting alone, would rather increase the snow albedo. Peltoniemi (2007) also report a slight darkening of snow with increasing density, which he ascribes to snow grains forming larger aggregates in a dense snowpack. Thus, the impacts of close-packing and effective snow grain size are intertwined.

## 8  Summary

In this work, the sensitivity of simulated climate to the assumed snow grain shape was studied using the NorESM model in a slab ocean configuration. Two snow grain shape assumptions were considered: spherical snow grains (the default assumption in NorESM) and non-spherical snow grains based on the optimized habit combination of Räisänen et al. (2015). The major findings were as follows:

- Snow albedo is higher when non-spherical rather than spherical snow grains are assumed, typically by 0.02–0.03, which results from a lower asymmetry parameter in the non-spherical case. The albedo difference gives rise to a global-mean instantaneous net shortwave flux difference (which may be thought as a "radiative forcing"; RF) of $-0.22\,\mathrm{W\,m^{-2}}$ at the top of the model atmosphere.

- In spite of the rather small global-mean RF, there are substantial climatic differences between the experiments with the non-spherical (NONSPH) and spherical (SPH) snow grains. The global annual-mean 2-m air temperature is $1.17\,\mathrm{K}$ lower in the NONSPH experiment than in the SPH experiment, with substantially larger differences at high latitudes. Furthermore, a northward shift of the ITCZ occurs in NONSPH. The climatic response is amplified by strong snow and sea ice feedbacks.

- The impact of changing the snow grain shape assumption in a climate model is largely equivalent to a change in the effective snow grain size $r_e$. Specifically, when the non-spherical snow grain shape assumption was used but the parameterized snow grain size was increased by ca. 70%, the climatic differences to the SPH experiment were very small.

- For given snowpack properties (i.e., SWE and $r_e$), the radiative effect due to absorbing aerosols in snow is generally smaller for non-spherical snow grains than for spherical snow grains, especially for a thick snowpack. The reason for this is that in the case of non-spherical snow grains, solar radiation does not penetrate as deep in the snowpack as in the case of spherical snow grains. However, in a climate model simulation, the annual-mean radiative effect due to absorbing aerosols in snow also depends on other factors, in particular how long snow persists on ground in spring. In fact, in the NONSPH experiment, the global land-area mean radiative effect due to absorbing aerosols in snow was 14% larger than in SPH, owing to delayed snowmelt in spring.

While real-world snow grains are definitely non-spherical rather than spherical in shape, the use of non-spherical snow grains in NorESM led to increased climatic biases, including a pronounced cold bias at high latitudes and too extensive snow and sea ice cover in summer. As such, this is not surprising, since parameterization changes very often lead to some deterioration of the





simulated climate, if not accompanied by model retuning. Perhaps of more concern, comparisons of model-simulated albedos with satellite surface albedo products and reflected solar radiation at the top of the atmosphere over the permanently snow-covered interior regions of Antarctica and Greenland suggested that the NONSPH experiment overestimates snow albedo. There are several uncertainty factors in the computation of snow albedo that could contribute to the overestimate in NONSPH.

- It is not clear how well the current prescription of snow single-scattering properties (derived from measurements of
angular scattering for two cases of blowing snow) represents snow in general. Specifically, if the typical asymmetry parameter of snow grains were larger than assumed here ($g \approx 0.78$ at weakly absorbing wavelengths), this would tend to make the NONSPH approach to overestimate snow albedo.

- Currently, the snow grain size is a poorly observed parameter, and therefore, the effective grain sizes simulated by NorESM must be considered uncertain. Specifically, should the simulated snow grain sizes be too small, this would
contribute to a high bias in simulated snow albedo.

- Other factors may also contribute, such as uncertainties in the effect of absorbing aerosols and the approximations made in the radiative transfer calculations. The latter include the use of the two-stream approximations instead of a numerically exact technique, the neglect of snow surface structure, and the neglect of close-packed effects.

  The primary conclusion from this work is that especially at high latitudes, the simulated climate can be quite sensitive to
the assumed snow grain shapes, and most probably, to the parameterization of snow albedo in general. To constrain snow albedo parameterizations better, more research is needed into observational evaluation of single-scattering properties of snow, the effective snow grain sizes, and the snow albedo itself.

*Code and data availability.* The NorESM model code is available for registered users. To register for access, users should contact noresm-ncc@met.mo and briefly state the purpose of the use of the model and sign a user agreement. In particular, the user agreement includes a
demand that all users of NorESM must register themselves as CESM users at the CESM website. Subroutines and datafiles needed for using optical properties for non-spherical snow grains are available at https://github.com/praisanen/snow_ssp, see "files_for_NorESM_or_CESM". Output data from the NorESM experiments can be obtained by contacting the first author.

## Appendix A: Derivation of snow single-scattering properties

  In the SNICAR model, the single scattering properties (SSPs) of snow (mass extinction coefficient $k_{\text{ext}}$, single-scattering
albedo $\omega$ and asymmetry parameter $g$) are tabulated for five spectral intervals (0.2–0.7, 0.7–1.0, 1.0–1.2, 1.2–1.5 and 1.5–5.0 $\mu$m) and for 1471 snow grain effective radii $r_e$ ranging from 30 to 1500 $\mu$m, separately for direct and diffuse solar radiation. Correspondingly, the ice shortwave radiation scheme in CICE4 defines the snow SSPs for three spectral intervals (0.2–0.7, 0.7–1.19, and 1.19–5.0 $\mu$m) for 32 values of $r_e$ ranging from 5 to 2500 $\mu$m (direct and diffuse radiation not separated).





We recomputed the snow SSPs in SNICAR and in CICE4, for consistency both for spheres and for the optimized habit
combination (Räisänen et al., 2015). Using the same single-scattering data of snow grains as in Räisänen et al. (2015), we first
computed, for each $r_e$ considered, high spectral resolution ($\Delta\lambda = 0.01\,\mu m$) snow SSPs integrated over a size distribution. A
lognormal size distribution with a geometric standard deviation of 1.5 was assumed.

Second, the high spectral resolution snow SSPs were averaged over each of the spectral bands in SNICAR and CICE4. The
extinction coefficient and asymmetry parameter were averaged using the standard approach (e.g., Slingo and Schrecker, 1982)

$$\overline{k_{\text{ext}}} = \frac{\sum_i S_i k_{\text{ext},i}}{\sum_i S_i} \tag{A1}$$

$$\overline{g} = \frac{\sum_i S_i k_{\text{ext},i} \omega_i g_i}{\sum_i S_i k_{\text{ext},i} \omega_i} \tag{A2}$$

Here, $k_{\text{ext},i}$, $\omega_i$ and $g_i$ are the extinction coefficient, single-scattering albedo and asymmetry parameter for the high-resolution
spectral intervals $i$ within the SNICAR or CICE4 spectral band considered, and $S_i$ is the downwelling spectral solar flux at
the surface, normalized so that its integral over the solar spectrum equals unity. Following Flanner et al. (2007), downwelling
solar spectrum for a cloud-free (overcast) midlatitude winter atmospere is used for averaging the SSPs in SNICAR for direct
(diffuse) solar radiation. For CICE4, we used the average normalized $S_i$ spectra for these two cases.

Choosing average values of $\omega$ for the spectral bands of SNICAR and CICE4 requires more care, as $\omega$ varies much more
strongly within these bands than $k_{\text{ext},i}$ and $g_i$ do. An approach similar to Eqs. (A1) and (A2) (weighting $\omega$ by $S_i k_{\text{ext},i}$)
would lead to a substantial underestimate of snow albedo and overestimated absorption in snow. Instead, we used an approach
similar to Briegleb and Light (2007): for each spectral band and and each value of $r_e$, the band-mean $\omega$ is adjusted so that the
band-mean snow albedo equals that obtained in high spectral-resolution ($\Delta\lambda = 0.01\,\mu m$) radiative transfer calculations for a
reference case:

$$R\left(\overline{k_{\text{ext}}}, \overline{g}, \overline{\omega}\right) = \frac{\sum_i S_i R_i\left(k_{\text{ext},i}, g_i, \omega_i\right)}{\sum_i S_i}, \tag{A3}$$

where $R$ is the snow albedo computed using the delta-Eddington method, and the summation is over the high-resolution spectral
intervals $i$ within the SNICAR or CICE4 spectral band considered. As the reference case, a vertically homogeneous snowpack
with SWE=$100\,\text{kg m}^{-2}$ and direct incident radiation with a zenith angle of $60°$ was assumed.

It should be noted that the SSPs derived for spheres deviate slightly from the original lookup tables in SNICAR and CICE,
both due to differences in details of spectral averaging and in the ice refractive index assumed. The original lookup tables are
based on the refractive index of Warren (1984) while here, newer data from Warren and Brandt (2008) are used. In particular,
in the newer data, the imaginary part of ice refractive index is substantially smaller in parts of the visible region. Nevertheless,
the snow albedo differences associated with the differences between the original and recomputed SSPs for spheres are smaller
by an order of magnitude than the corresponding differences between spheres and the OHC.

*Author contributions.* PR designed the study, run the NorESM experiments, analyzed the results and wrote the manuscript. RM, AK and JD
provided scientific and technical advice regarding NorESM and commented on the manuscript.





*Competing interests.*  The authors declare that they have no conflict of interest.

*Acknowledgements.*  P. Räisänen acknowledges funding by the Academy of Finland through the NABCEA project (decision number 296302). A. Kirkevåg and J. B. Debernard have been funded by the Research Council of Norway through the EarthClim (20771/E10), EVA(229771), NOTUR(nn2345k) and Norstore (ns2345k) projects, by the Nordic Centers of Excellence CRAICC and eSTICC, and by the EU FP7 projects PEGASOS and ACCESS. Mark Flanner is thanked for providing the spectral distribution of downwelling solar radiation for midlatitude

winter conditions. We also wish to thank the data providers: ECMWF for the ERA-Interim data, NOAA for the snow cover and sea ice climate data records, NASA Langley Research Center Atmospheric Science Data Center for the MODIS and CERES EBAF data, and EUMETSAT Satellite Application Facility on Climate Monitoring (CM SAF) for the CLARA-A2 data.





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





**Table 1.** List of experiments conducted. The columns 2 through 5 give snow grain shape on land and sea ice, the grain size parameterization and atmospheric $CO_2$ concentration (ppmv). "OHC" refers to the optimized habit combination (Räisänen et al., 2015) and "default" to the default parameterization of snow grain size in SNICAR and CICE4.

| experiment | grain shape on land | grain shape on ice | grain size | $CO_2$ |
|---|---|---|---|---|
| SPH | spheres | spheres | default | 367 |
| NONSPH | OHC | OHC | default | 367 |
| NONSPHLND | OHC | spheres | default | 367 |
| NONSPHICE | spheres | OHC | default | 367 |
| TUNED | OHC | OHC | increased by ~70% | 367 |
| SPH2XCO2 | spheres | spheres | default | 734 |



**Table 2.** Annual-mean reflected shortwave radiation ($\mathrm{W\,m^{-2}}$) at the top of the atmosphere, averaged over the Central Antarctica and Central Greenland regions shown in Fig. 10 for the CERES EBAF Ed2.8 dataset and for the SPH and NONSPH experiments. The differences from the CERES EBAF data are given in parentheses.

|  | CERES | SPH | NONSPH |
|---|---|---|---|
| Central Antarctica, all-sky | 122.8 | 126.4 (3.6) | 131.0 (8.2) |
| Central Antarctica, clear-sky | 121.1 | 124.3 (3.2) | 129.8 (8.8) |
| Central Greenland, all-sky | 130.1 | 130.5 (0.3) | 133.7 (3.6) |
| Central Greenland, clear-sky | 126.9 | 125.4 (−1.5) | 130.2 (3.3) |





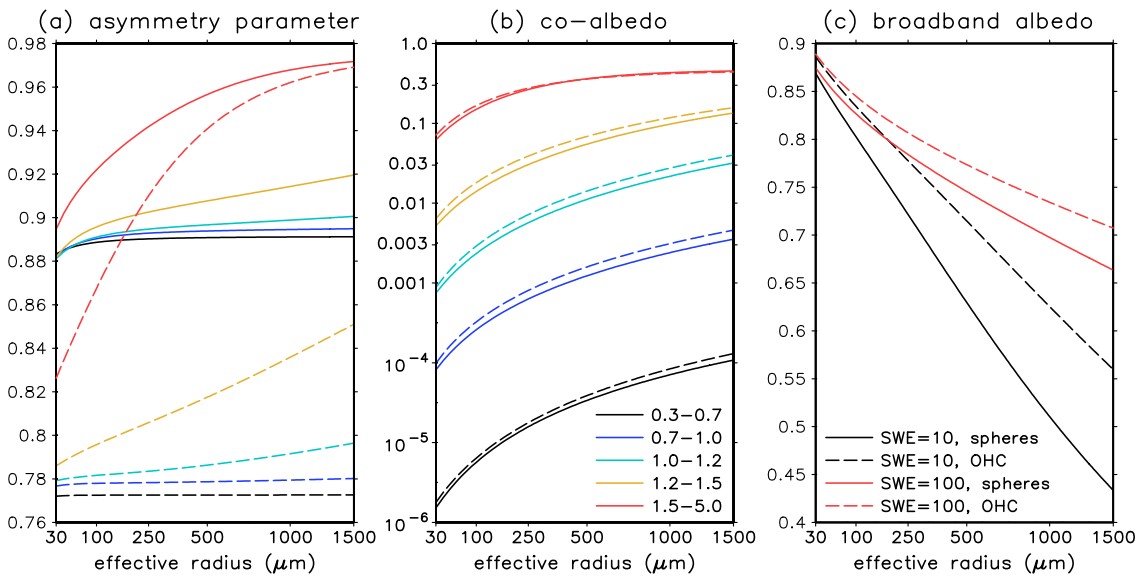

**Figure 1. (a)** Asymmetry parameter and **(b)** single-scattering co-albedo of snow grains for the five spectral intervals of SNICAR for spheres (solid lines) and for the OHC (dashed lines) as a function of snow grain effective radius. See the legend in panel **(b)** for the intervals (in $\mu$m). **(c)** Corresponding broadband (0.2–5.0 $\mu$m) albedo of pure snow, for snow water equivalents of SWE=10 kg m$^{-2}$ (black lines) and SWE=100 kg m$^{-2}$ (red lines). Snow albedo was computed using the delta-Eddington method in the visible region and the delta-hemispheric mean method in the near-IR region assuming an underlying surface albedo of 0 and direct incident solar radiation with a zenith angle of $60°$ and with a spectral distribution corresponding to a cloud-free midlatitude winter atmosphere.





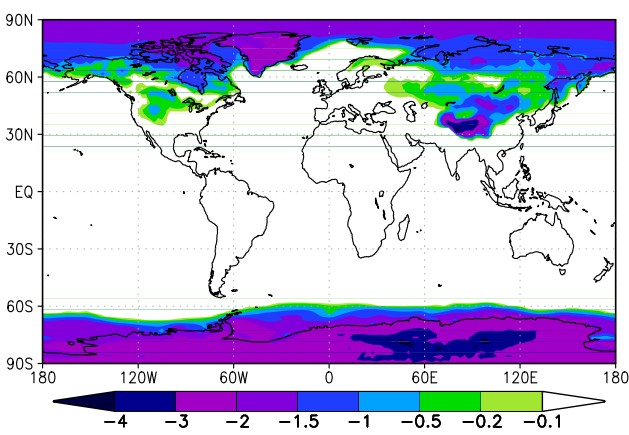

**Figure 2.** Annual-mean instantaneous radiative effect at the top of the model atmosphere ($\mathrm{W\,m^{-2}}$) due to changing the snow grain shape assumption from spheres to the OHC, evaluated through diagnostic radiation calculations in the SPH experiment.





**Figure 3.** Annual-mean differences in **(a)** 2-m air temperature and **(b)** in precipitation between the NONSPH and SPH experiments; **(c–d)** between the NONSPHLND and SPH experiments; and **(e–f)** between the NONSPHICE and SPH experiments. Global mean differences are given in the panel titles. Hatching indicates differences that exceed the limit for colour shading (0.2 K for temperature and 0.1 mm d$^{-1}$ for precipitation) but are not significant at the 5% level.



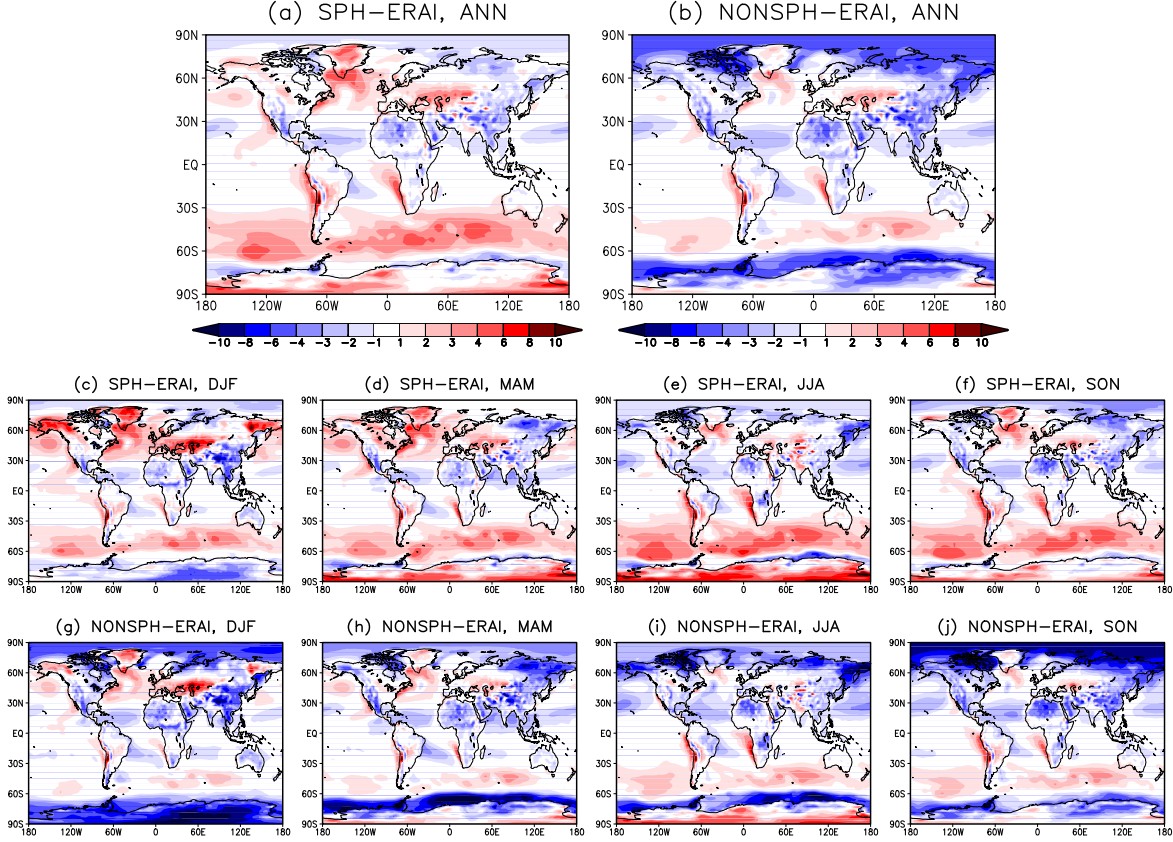

**Figure 4.** Annual-mean differences in 2-m air temperature **(a)** between the SPH experiment and ERA-Interim reanalysis (averaged over the years 1990–2014) and **(b)** between NONSPH and ERA-Interim. **(c)–(f)** Corresponding seasonal-mean differences between SPH and ERA-Interim, and **(g)–(j)** between NONSPH and ERA-Interim.





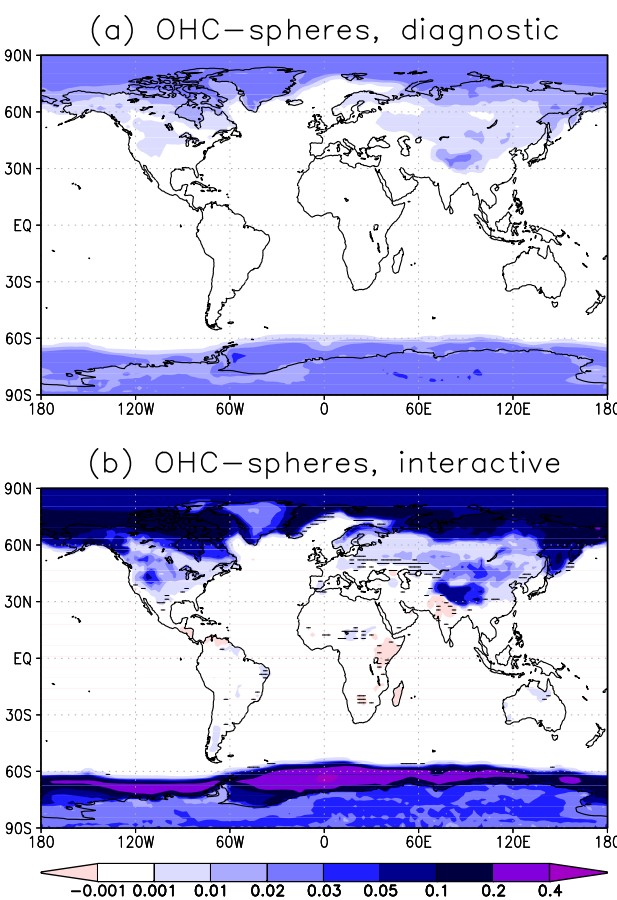

**Figure 5. (a)** Difference in annual-mean surface albedo between the OHC and spherical snow grain shape assumptions, based on diagnostic radiation calculations in the SPH experiment. **(b)** The actual surface albedo difference between the NONSPH and SPH experiments, in which the OHC and spherical shape assumptions, respectively, influence the climate simulation interactively. The albedos are weighted by the incoming solar flux at the surface. Hatching indicates differences that exceed the limit for colour shading (0.001) but are not significant at the 5% level.





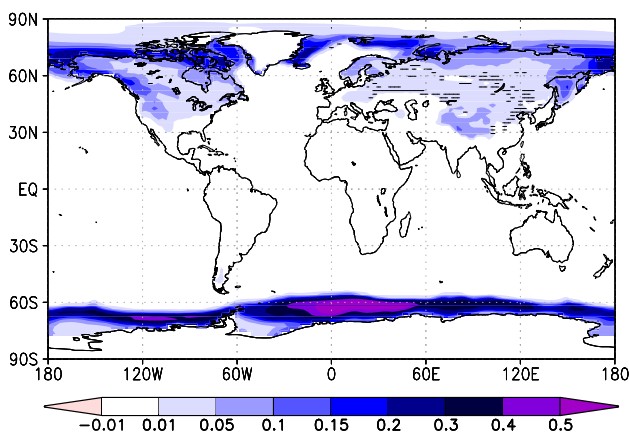

**Figure 6.** Differences between the NONSPH and SPH experiments in annual-mean snow fraction on land and in sea ice cover. Hatching indicates differences that exceed the limit for colour shading (0.01) but are not significant at the 5% level.



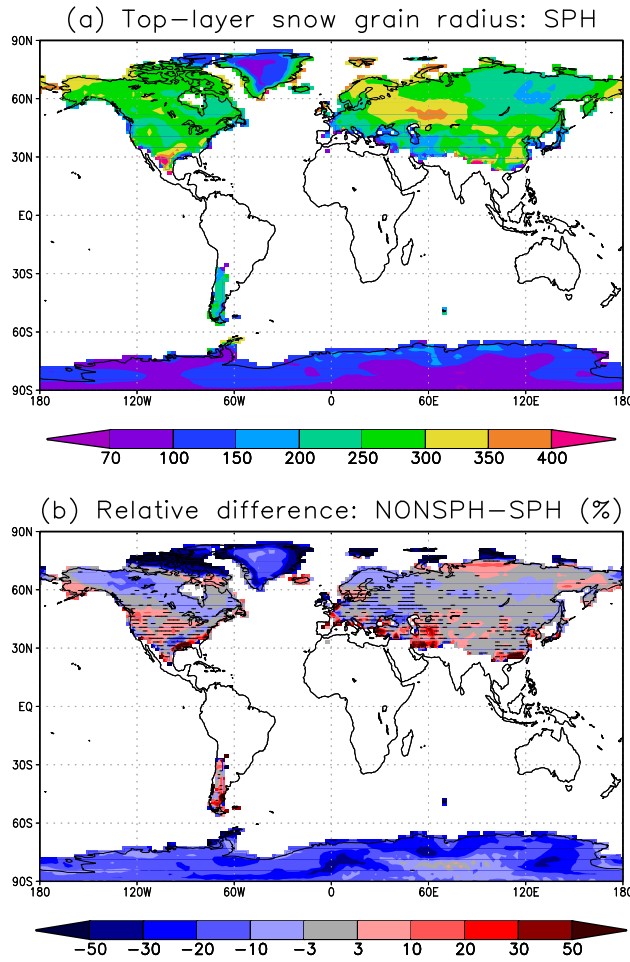

**Figure 7. (a)** Mean value of snow grain effective radius in the uppermost snow layer on land in the SPH experiment (µ m) (weighted by snow cover fraction). **(b)** The corresponding relative difference (in %) between the NONSPH and SPH experiments. Hatching indicates differences that exceed the limit for colour shading (3 %) but are not significant at the 5% level.



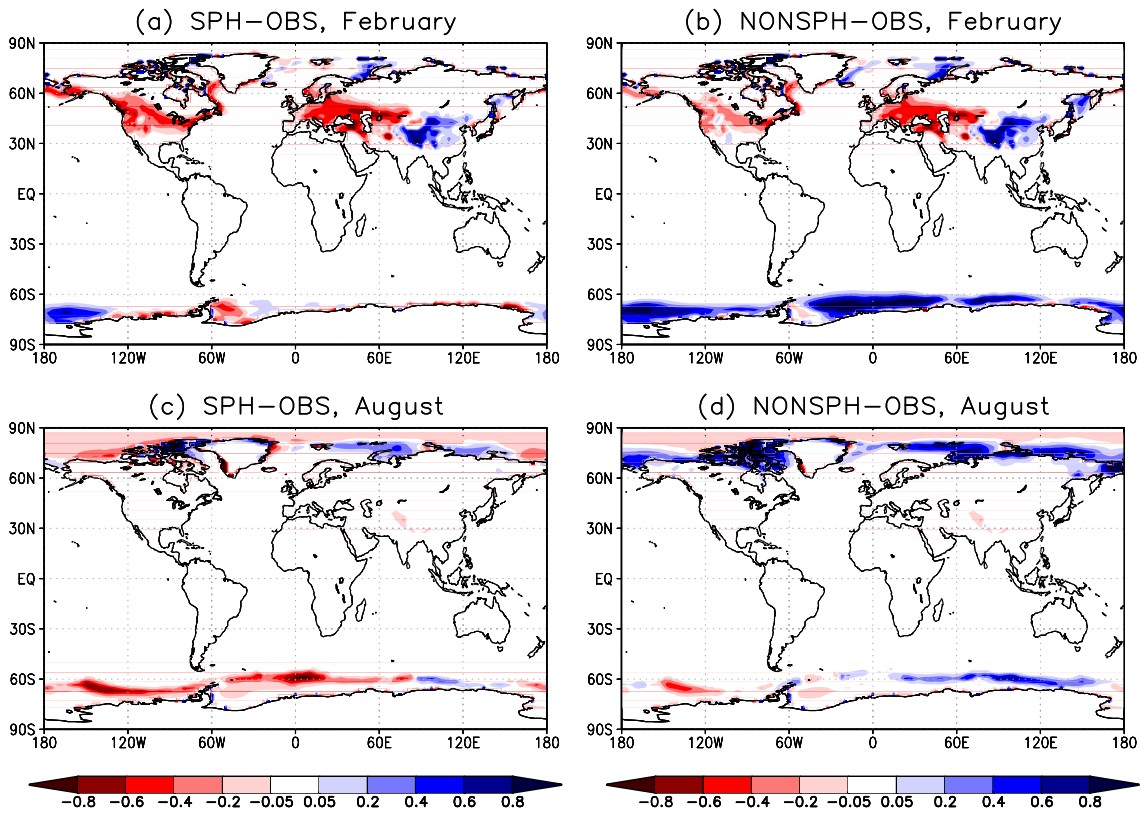

**Figure 8.** Differences in snow cover on land and in sea ice cover between the SPH and NONSPH experiments and observations (OBS), for the months of **(a,b)** February and **(c,d)** August. The observations for land snow cover are from the NOAA northern hemisphere snow cover extent CDR (Robinson et al., 2012; Estilow et al., 2015) and the sea ice observations from the NSIDC passive microwave sea ice concentration CDR (Meier et al., 2013; Peng et al., 2013), both averaged over 1990–2014.





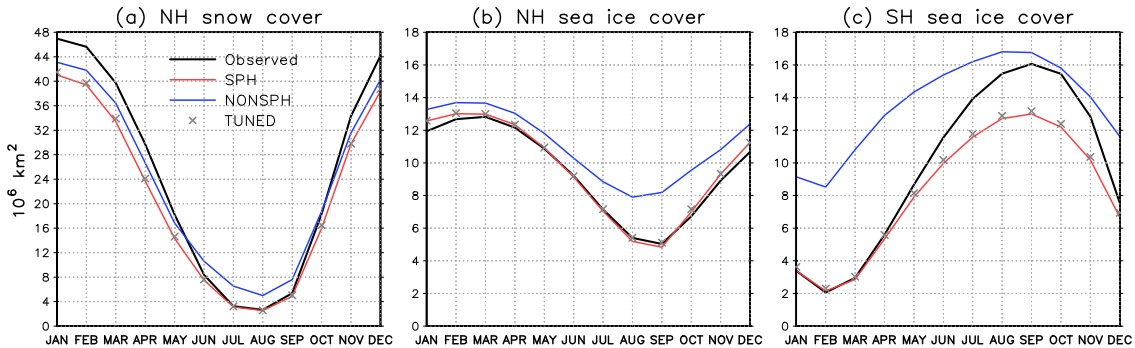

**Figure 9. (a)** Seasonal cycle of northern hemisphere land snow cover in the NOAA snow cover extent CDR (Robinson et al., 2012; Estilow et al., 2015) (black curve) and in the SPH (red curve), NONSPH (blue curve) and TUNED (crosses) experiments. **(b)** Northern hemisphere and **(c)** southern hemisphere sea ice cover in the NSIDC passive microwave sea ice concentration CDR (Meier et al., 2013; Peng et al., 2013) and in the NorESM experiments. For both snow cover and sea ice, the observations are averaged over the years 1990–2014. To account for missing sea ice data in the NSIDC product around the North pole, $0.33 \times 10^6$ km$^2$ has been added to the observed values in **(b)**.



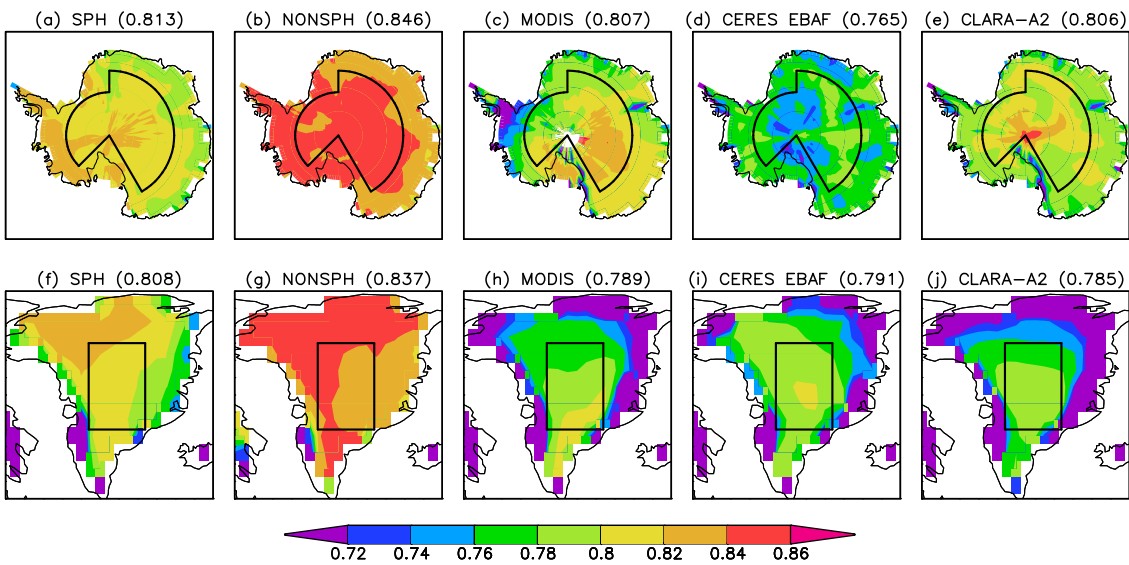

**Figure 10.** Annual-mean surface albedo **(a–e)** over Antarctica and **(f–j)** over Greenland in the SPH and NONSPH experiments and in three observational datasets: MODIS, CERES EBAF and CLARA-A2. Clear-sky albedos are shown for SPH, NONSPH and CERES EBAF, and black-sky albedos for MODIS and CLARA-A2. The numerical values given in parentheses are mean albedos averaged over the two regions marked in the figures: Central Antarctica (75–90° S at 0–150° E plus 80–90° S at 0–135° W) and Central Greenland (68–78° N, 33–48° W). In temporal and spatial averaging, the albedo values are weighted by the clear-sky downwelling solar flux at the surface, using modelled values for the SPH and NONSPH experiments and CERES EBAF values for the three satellite datasets.



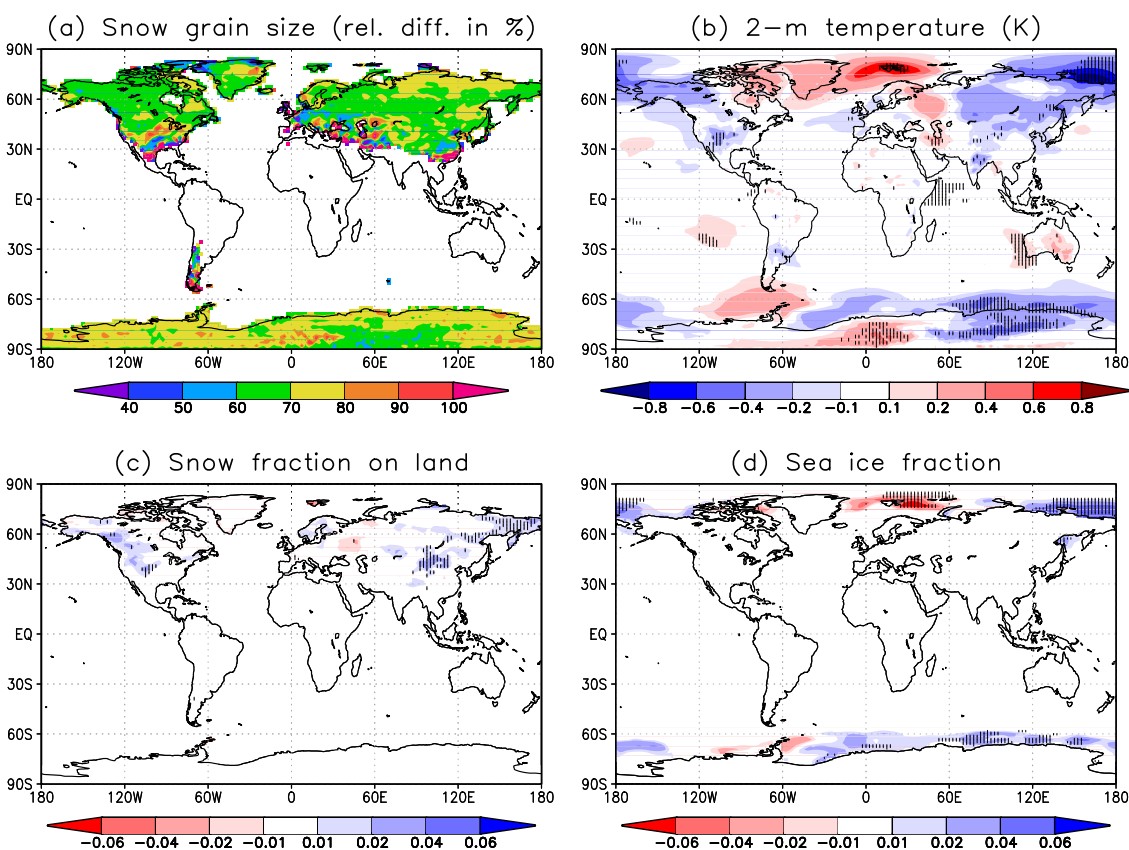

**Figure 11.** Annual-mean differences between the TUNED and SPH experiments: **(a)** the relative difference in the snow grain effective radius in the uppermost snow layer on land (in %), **(b)** the difference in 2-m air temperature (K), **(c)** the difference in snow fraction on land, and **(d)** the difference in sea ice fraction. In **(b)–(d)**, differences significant at the 5% level are indicated with vertical hatching.





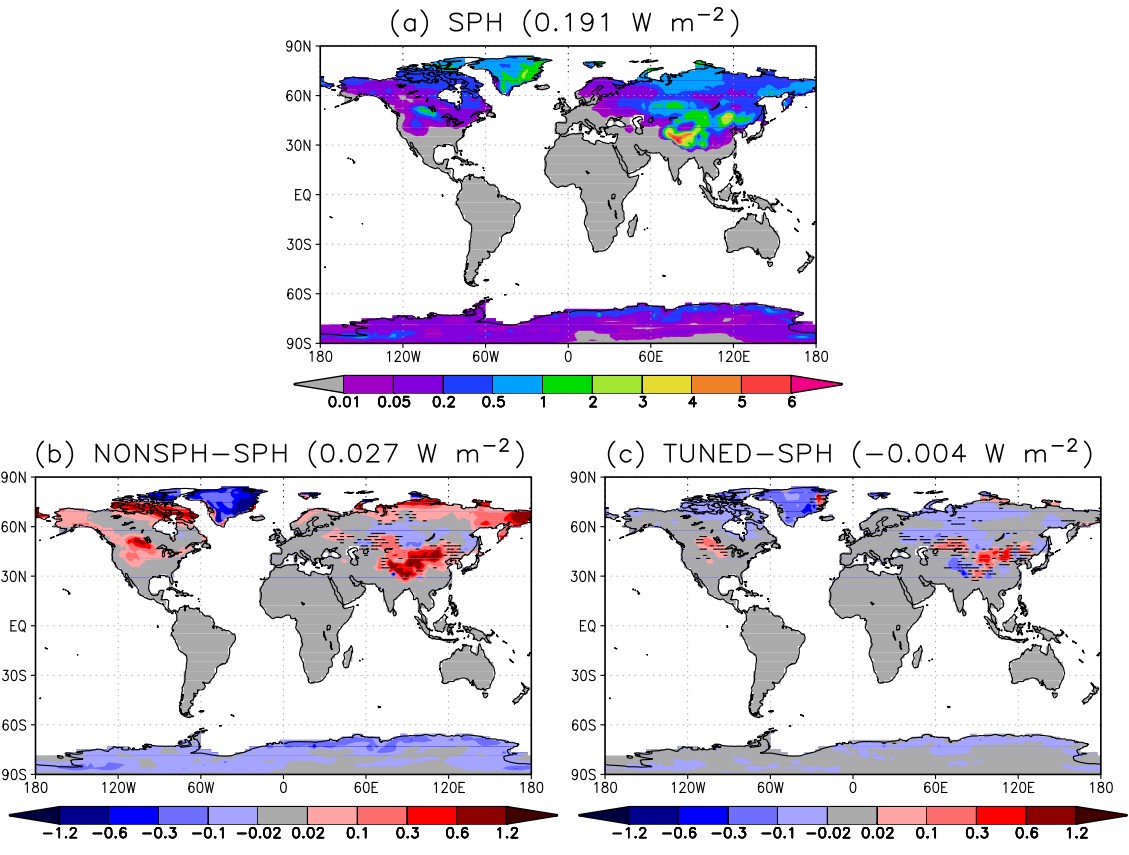

**Figure 12.** **(a)** Annual-mean surface radiative effect of absorbing aerosols in snow on land in the SPH experiment ($\mathrm{W\,m^{-2}}$). **(b)** The corresponding differences between the NONSPH and SPH experiments, and **(c)** between the TUNED and SPH experiments. Global land-area mean values are given in the panel titles. Hatching indicates differences that exceed the limit for colour shading ($0.02\,\mathrm{W\,m^{-2}}$) but are not significant at the 5% level.





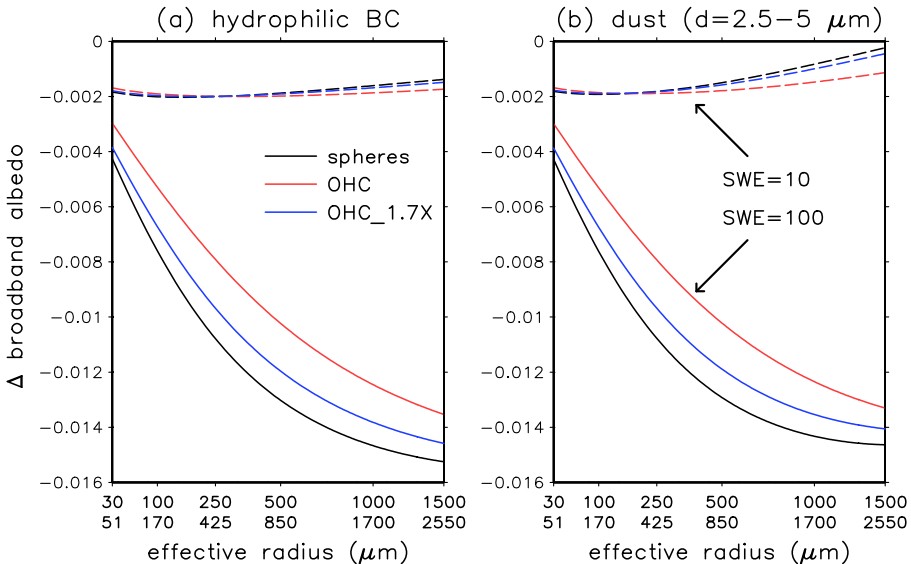

**Figure 13.** Impact of absorbing aerosols on snow broadband albedo as a function of snow grain effective radius $r_e$ for **(a)** hydrophilic BC and **(b)** mineral dust (second-largest dust size bin considered in SNICAR, corresponding to particle diameters of 2.5–5 µm). The solid lines are for a snow water equivalent SWE=100 kg m$^{-2}$ and the dashed lines for SWE=10 kg m$^{-2}$, while the black, red, and blue colours correspond to spheres, the OHC, and the OHC with $r_e$ multiplied by 1.7, respectively. The upper scale on the $x$-axis gives $r_e$ for the former two cases and the lower scale for the last case. The results were computed using the delta-Eddington method in the visible region and the delta-hemispheric mean method in the near-infrared, assuming direct incident solar radiation with a zenith angle of 60° and with a spectral distribution corresponding to a cloud-free midlatitude winter atmosphere, with an underlying surface albedo of 0. The aerosols are homogeneously mixed in snow, with mass-mixing ratios selected so that for both aerosol species, the albedo reduction equals 0.01 for spherical snow grains with $r_e$=200 µm when SWE=100 kg m$^{-2}$ (23.9 ppb for hydrophilic BC and 8.12 ppm for dust).