# Peer review of "Effects of snow grain shape on climate simulations: Sensitivity tests with the Norwegian Earth System Model"

_The Cryosphere, 2017_

## Referee Comment (RC1) · Anonymous Referee #1 · 13 Sep 2017

Recommendation: Accept after minor revision. This is an interesting study which is suitable for publication in The Cryosphere. It is well-written.

lines 8-9. Say that the nonspherical grains are compared to spherical grains with the same specific surface area.

line 32. Cite also Dang et al. (2015).

line 47. If the snow grain contains concavities and hollows, then the projected area is not the appropriate measure, because internal surfaces also deflect photons. See Grenfell et al. (2005). Admittedly, although cavities are present in atmospheric ice crystals, they are uncommon in surface snow.

line 49, eq. 1. Point out that re is inversely proportional to specific surface area (SSA), a quantity that is commonly used in snow radiation work.

line 89. Change "retuning the snow grain size" to "increasing the snow grain size (of the nonspherical grains)"

line 96. "model model" is redundant.

line 183. "abundant snow cover . . . in parts of Tibet . . . " What does NorESM predict for snow cover and snow depth in Tibet? In reality, Tibetan snow is patchy and thin, with average depth peaking in February at only 2 cm (Flanner and Zender 2005, Figure 3b).

line 185-186. "in the southern parts of northern Eurasia . . . the change in snow albedo is largely masked by forests." This is also seen in a band of forest across North America at 50-60N between the Great Plains and the tundra.

line 209-210. Define "Q-flux".

line 296. Change "Figs. 8 and Fig. 9" to "Figs. 8 and 9".

line 303. Change "or" to "of".

line 357. Change "NONSPH" TO "SPH". This is important.

line 372-373. "2 W m-2 in eastern Greenland (mainly due to BC)". This is probably excessive. The BC content at East Greenland AWS stations is only 2-4 ppb (Table 6 of Doherty et al. 2010).

line 441. Change "in lack of information" to "because of the lack of information".

line 447. "indistinguishable" is misspelled.

line 516. Change "report" to "reports".

line 603. Change "run" to "ran".

Figure 3 caption line 3. Change "limit" to "threshold". Also on captions to Figures 5 and 6.

Figure 7a. Give units on scale bar (probably micrometers).

Figure 7b. A ratio (rather than percent-difference) might be esier for the reader to interpret. Also in Figure 11a.

Figure 13 caption last line. These numbers will be easier to compare if they are given in the same units: "(24 ppb for hydrophilic BC and 8120 ppb for dust)".

References

Dang, C., R.E. Brandt, and S.G. Warren, 2015: Parameterizations for narrowband and broadband albedo of pure snow, and snow containing mineral dust and black carbon. J. Geophys. Res., 120, doi:10.1002/2014JD022646.

Doherty, S.J., S.G. Warren, T.C. Grenfell, A.D. Clarke, and R.E. Brandt, 2010: Light-absorbing impurities in Arctic snow. Atmos. Chem. Phys., 10, 11647-11680.

Flanner, M.G., and C.S. Zender, 2005: Snowpack radiative heating: Influence on Tibetan Plateau climate. Geophys. Res. Lett., 32, L06501, doi:10.1029/2004GL022076.

Grenfell, T.C., S.P. Neshyba, and S.G. Warren, 2005: Representation of a nonspherical ice particle by a collection of independent spheres for scattering and absorption of radiation: 3. Hollow columns and plates. J. Geophys. Res. (Atmospheres), 110, D17203, doi:10.1029/2005JD005811.

---

## Referee Comment (RC2) · Anonymous Referee #2 · 10 Oct 2017

The study investigates the impacts of including the effects of non-sphericity of snow grains in the NorESM model in slab ocean mode. They find that due to a smaller asymmetry parameter, snow broadband albedo is generally higher. Despite a global mean radiative forcing associated with this change of only -0.22 W/m2, this leads to considerable differences in the simulated equilibrium climate – particularly at high latitudes.

The paper is very well written, structured, illustrated and argued. It is, quite frankly, a pleasure to read and I have only extremely minor suggestions for improvement. It fits within the journal and I suggest to accept with only minor changes.

L109: Would it be relevant to mention (just briefly) if and how the sea ice model is dynamic and how this works on top of the slab ocean?

Eqs (3) and (4): Perhaps note that the time step DeltaT is multiplied onto these rates to get the dr's used in Eqn (2).

L226: at the end of the sentence, do you mean "SPH and ERA-Interim reach -7 K"?

L246: Perhaps note that these diagnostic calculations were the same used to calculate the TOM RF.

L264: "smaller contribution". It is not completely clear to me how you see that this contribution is smaller. Please explain better.

L322-330+Table 2: Is this part really necessary? If you want to shorten, this could be a place.

L337: yes, because the default parameterization was used when other parts of the model were tuned originally.

L357:NONSPH->SPH

L516: reports

L529: Perhaps: "at high latitudes up to XX K."

L531: Perhaps add to end of sentence something like "and therefore leads to large efficacy of the RF". This would better justify including the efficacy section in Section 7.

L552: Delete "to"

---

## Short Comment (SC1) · 18 Oct 2017

The authors conducted a series of sensitivity simulations using the Norwegian Earth System Model to quantify the effects of snow grain shape, which could improve the understanding on the role of snow grain shape in climate modeling. For the authors' information, a very recent study (He et al., 2017) did a detailed analysis and parameterization to account for snow grain shape effects on optical properties of both clean and dirty snow, which could be cited and discussed to improve the discussions in the manuscript.

Reference: He, C., Y. Takano, K. Liou, P. Yang, Q. Li, and F. Chen, 2017: Impact of

[Figure]

Snow Grain Shape and Black Carbon-Snow Internal Mixing on Snow Optical Properties: Parameterizations for Climate Models. J. Climate, 0, https://doi.org/10.1175/JCLI-D-17-0300.1

---

## Author Comment (AC1) · 30 Oct 2017

We thank Anonymous Referee #1 for his/her constructive comments on the manuscript. Point-by-point responses to the comments are provided below. The referee comments are written in *italic* font, and our responses in normal font.

**Comment:** Recommendation: Accept after minor revision. This is an interesting study which is suitable for publication in The Cryosphere. It is well-written.

lines 8-9. Say that the nonspherical grains are compared to spherical grains with the same specific surface area.

**Response:** The effective radius is defined in our study using the ratio of volume to projected area rather than volume to total surface area. Therefore, even if the difference might in practice be small for snow, it is in principle more correct to use here "specific projected area" instead of "specific surface area".

**Change in the manuscript**: In the abstract, it will be said: "Therefore, for the same snow grain effective size (or equivalently, the same specific projected area), the snow broadband albedo is higher when assuming non-spherical rather than spherical snow grains, typically by 0.02–0.03."

Comment: line 32. Cite also Dang et al. (2015)

**Response and change:** We will cite this paper in the revised manuscript.

**Comment:** line 47. If the snow grain contains concavities and hollows, then the projected area is not the appropriate measure, because internal surfaces also deflect photons. See Grenfell et al. (2005). Admittedly, although cavities are present in atmospheric ice crystals, they are uncommon in surface snow.

**Response:** In fact, we agree only partially with this comment. It is certainly true that the single-scattering properties of non-spherical particles can be influenced by concavities and hollows, and it is also true that no definition of effective radius (including our Eq. 1) is *always* optimal. However, it is not at all clear that defining the effective radius in terms
of the total surface area A

$$r_{VA} = 3\frac{V}{A} \tag{1}$$

rather than in terms of the projected area P (which is what we use in the manuscript, although with the notation  $r_e$ )

$$r_{VP} = \frac{3V}{4P} \tag{2}$$

would be an improvement, even in the case of concave particles. It should be noted that Grenfell et al. (2005) only consider  $r_{VA}$  and do not compare its performance to  $r_{VP}$ , so their paper does not yield direct information on which choice actually works better. However, for concave particles  $r_{VA}

**Change in the manuscript:** The following will be stated in the revised manuscript: "While the SSPs of non-spherical particles (including snow grains) can be influenced by concavities and hollows (Grenfell et al. 2005), in general, the most relevant measure of their size for radiative transfer is the volume-to-projected area equivalent effective radius (e.g., Mitchell 2002) ..."

**Comment:** line 49, eq. 1. Point out that re is inversely proportional to specific surface area (SSA), a quantity that is commonly used in snow radiation work.

**Response:** The  $r_e$  is inversely proportional both to SSA and to the specific projected area (SPA). For completeness, it is perhaps best to add an equation showing these relationships.

**Change in the manuscript:** We will note the following: "The  $r_e$  is inversely proportional to the snow specific projected area (SPA; projected area per mass) and the specific surface area (SSA; total surface area per mass):

$$r_e = \frac{3}{4\rho_{\rm ice} {\rm SPA}} = \frac{3F}{\rho_{\rm ice} {\rm SSA}} \tag{3}$$

where  $\rho_{ice}$  is the density of ice and the fluffiness parameter F = SSA/4SPA (Grenfell et al. 2005) is F = 1 for convex particles such as spheres and F > 1 for concave particles."

**Comment:** *line 89. Change "retuning the snow grain size" to "increasing the snow grain size (of the nonspherical grains)"*

**Response and change:** This will be reworded according to the suggestion.
Response and change: This will be corrected.

**Comment:** line 183. "abundant snow cover ... in parts of Tibet ...". What does NorESM predict for snow cover and snow depth in Tibet? In reality, Tibetan snow is patchy and thin, with average depth peaking in February at only 2 cm (Flanner and Zender 2005, Figure 3b).

**Response:** It is indeed true that NorESM overestimates the amount of snow in Tibet. The snow cover fraction in February is close to 80%, as compared with 30% in the NOAA SCE CDR (aka. Rutgers University) data, resulting in the distinct overestimate seen in Fig. 8a-b. The area-mean snow depth in February for the region considered by Flanner and Zender (2005) is about 25 cm in SPH and 31 cm in NONSPH. Incidentally, while overestimated snow cover likely exaggerates the "radiative forcing" associated with changed snow grain size, it is not obvious that overestimated snow depth works in the same direction. In fact, Fig. 1c of our manuscript indicates that snow grain shape has a larger effect on snow broadband albedo when the snow layer is relatively thin.

**Change in the manuscript:** We will add the following in Sect. 4.1 (as a footnote, to avoid the disruption of the flow of the main text): "The RF in Tibet may be exaggerated by NorESM's overestimation of snow cover in Tibet (see Fig. 8a–b below)." In addition, the snow depth will be mentioned in Sect. 4.3.1 where snow-related quantities are compared with observations. "... overestimation in Tibet (Fig. 8a–b), where snow depth is also overestimated (the February mean snow depth for the Tibetan Plateau region (30–40°N, 80–100°E) being 25 cm for SPH and 31 cm for NONSPH, as
compared with roughly 10 cm for satellite microwave-derived data and only 2 cm for in situ data, see Fig. 3b in Flanner et al. (2005))".

**Comment:** line 185-186. "in the southern parts of northern Eurasia ... the change in snow albedo is largely masked by forests." This is also seen in a band of forest across North America at 50-60N between the Great Plains and the tundra.

**Response and change:** This is true and will be noted it in the revised manuscript.

Comment: line 209-210. Define "Q-flux".

**Response:** The physical meaning of Q-fluxes is explained in connection with the description of the mixed-layer ocean model in Sect. 2 (lines 105–106 of the original manuscript): "The Q-flux (representing the implied horizontal and vertical heat flux into/out of the local mixed-layer column)...", so there is no need to repeat this explanation in Sect 4.2.

**Change in the manuscript:** For clarity, we will add a reference to Sect. 2 at the point where *Q*-fluxes are mentioned in Sect 4.2. "... especially because the *Q*-fluxes employed in the slab ocean model are based on a preindustrial simulation (see Sect. 2)".

Comment: line 296. Change "Figs. 8 and Fig. 9" to "Figs. 8 and 9".

**Response and change:** This will be changed as suggested.

TCD
Comment: line 303. Change "or" to "of".

Response and change: This will be corrected.

Comment: line 357. Change "NONSPH" TO "SPH". This is important.

Response and change: Thanks for spotting this! It will be corrected.

**Comment:** *line 372-373. "2 W m-2 in eastern Greenland (mainly due to BC)". This is probably excessive. The BC content at East Greenland AWS stations is only 2-4 ppb (Table 6 of Doherty et al. 2010).*

**Response:** We compared simulated surface-layer BC concentrations in the SPH experiment to the observations listed in Table 6 of Doherty et al. (2010). This comparison indicated a slight overestimation for the spring measurements (simulated BC concentrations 2–10 ppb, observed 2–7 ppb) and a more pronounced overestimation for the summer measurements (simulated BC concentrations 7–23 ppb, observed 1–20 ppb but mostly 1–4 ppb). However, it may be noted that the summer measurements are mostly at different sites than the spring measurements, and in fact no measurements are available in the region where the NorESM aerosol radiative effect is maximum. In that region, the simulated BC concentrations in surface snow in summer were as high as  $\sim$ 30–40 ppb, which is very probably too much, even if some enrichment of BC in the surface snow is likely to happen during the snow melt season also in reality.

Change in the manuscript: In the interest of brevity, we will only add the following sentence regarding this issue in the revised manuscript: "The RE in Greenland may
be excessive, as comparison with observed BC concentrations in Greenland (Table 6 in Doherty et al. 2010) suggested that NorESM likely overestimates the BC in surface snow especially in summer."

**Comment:** line 441. Change "in lack of information" to "because of the lack of information".

Response and change: This will be corrected as suggested.

Comments: line 447. "indistinguishable" is misspelled.

line 516. Change "report" to "reports".

line 603. Change "run" to "ran".

Response and change: These typos will be corrected.

**Comment:** Figure 3 caption line 3. Change "limit" to "threshold". Also on captions to Figures 5 and 6.

**Response and change:** Yes, "threshold" is a better word here. This also applies to captions to Figs. 7 and 12. These will be corrected as suggested.

**Comment:** Figure 7a. Give units on scale bar (probably micrometers).
**Response and change:** We will add the units ( $\mu$ m) to the panel title of Fig. 7a (it is technically easier, and more consistent with our other figures).

**Comment:** Figure 7b. A ratio (rather than percent-difference) might be easier for the reader to interpret. Also in Figure 11a.

**Response and change:** We will modify these figures as suggested and update the wording accordingly.

**Comment:** Figure 13 caption last line. These numbers will be easier to compare if they are given in the same units: "(24 ppb for hydrophilic BC and 8120 ppb for dust)".

Response and change: This will be changed as suggested.

---

## Author Comment (AC2) · 30 Oct 2017

We thank Anonymous Referee #2 for his/her constructive comments on the manuscript. Point-by-point responses to the comments are provided below. The referee comments are written in *italic* font, and our responses in normal font.

**Comment:** *The study investigates the impacts of including the effects of non-sphericity of snow grains in the NorESM model in slab ocean mode. They find that due to a smaller asymmetry parameter, snow broadband albedo is generally higher. Despite a global mean radiative forcing associated with this change of only -0.22 W/m2, this*

[Figure]

*leads to considerable differences in the simulated equilibrium climate — particularly at high latitudes.*
*The paper is very well written, structured, illustrated and argued. It is, quite frankly, a pleasure to read and I have only extremely minor suggestions for improvement. It fits within the journal and I suggest to accept with only minor changes.*

*L109: Would it be relevant to mention (just briefly) if and how the sea ice model is dynamic and how this works on top of the slab ocean?*

**Response and change in the manuscript:** Yes, CICE4 is a dynamic sea ice model. We will add the following explanation: "Like fully coupled configurations, the slab ocean setup uses the full prognostic thermodynamic and dynamic configuration of the sea ice model. Ice velocities are prognostic and calculated based on winds from the atmosphere model and ocean currents specified from an earlier fully coupled run with NorESM (the same preindustrial control simulation as used for calculating the $Q$-fluxes for the slab ocean component). In addition to the thermodynamic response, this allows for transport and deformation of sea ice in response to changes in the atmospheric circulation."

**Comment:** *Eqs (3) and (4): Perhaps note that the time step DeltaT is multiplied onto these rates to get the dr's used in Eqn (2).*

**Response and change:** In the revised manuscript, we will include $\Delta t$ on the rhs of Eqs. (3) and (4), so they directly give the $dr_{e,\mathrm{dry}}$ and $dr_{e,\mathrm{wet}}$ needed in Eq. (2).

**Comment:** *L226: at the end of the sentence, do you mean "SPH and ERA-Interim reach -7 K"?*

**Response:** No, the sentence is correct as it is. Its purpose is to stress that NONSPH also differs a lot from SPH, not only ERA-Interim. (The maximum difference between SPH and ERA-Interim in Antarctica in summer is close to $-6\,\mathrm{K}$). Thus, no change in the manuscript.

**Comment:** *L246: Perhaps note that these diagnostic calculations were the same used to calculate the TOM RF.*

**Response and change:** This will be noted in the revised manuscript.

**Comment:** *L264: "smaller contribution". It is not completely clear to me how you see that this contribution is smaller. Please explain better.*

**Response:** Quantifying precisely how much the change in snow grain size $r_e$ contributes to the albedo difference between NONSPH and SPH would require extensive and laborious off-line radiation calculations. However, even without performing such calculations, it is safe to say that the contribution from changed $r_e$ to the overall albedo diffence seen in Fig. 5b is (much) smaller than the contribution from increased snow and sea ice cover. The impact of the change in $r_e$ can best be discerned from other factors in regions with permanent snow cover (i.e., Greenland and Antarctica); it is also these regions which show the most consistent decrease in $r_e$ in the NONSPH experiment in Fig. 7b. By taking the difference between the actual albedo difference between NONSPH and SPH (Fig. 5b) and the corresponding albedo difference from diagnostic radiation calculations (Fig. 5a), we estimate that the reduced $r_e$ in NONSPH increases the snow albedo in these regions on average by 0.007, although the difference reaches 0.02 for some grid points. Albedo differences of this size are

quite small compared to the differences between NONSPH and SPH in those regions where NONSPH features increased snow and sea ice cover (which can be well above 0.1; Fig. 5b). Therefore, beyond reasonable doubt, the dominant contribution to the albedo differences in Fig. 5b comes from changed snow and sea ice cover rather than changed $r_e$.

**Change in the manuscript:** In the revised manuscript, we will include a shorter version of the above explanation at the end of Sect. 4.3.: "The impact of this is most easily discernible in the permanently snow-covered regions of Antarctica and Greenland. In these regions, the actual albedo difference between NONSPH and SPH (Fig. 5b) is slightly larger than that derived from diagnostic radiation calculations (Fig. 5a); on average by 0.007 and by up to 0.02 for some grid points. These albedo changes are, however, much smaller than the albedo differences between NONSPH and SPH in regions of changed snow and sea ice cover."

**Comment:** *L322-330+Table 2: Is this part really necessary? If you want to shorten, this could be a place.*

**Response and change:** True enough, it is not strictly necessary. This part will be eliminated in the revised manuscript.

**Comment:** *L337: yes, because the default parameterization was used when other parts of the model were tuned originally.*

**Response and change:** The following sentence will be added to the revised manuscript: "In particular, the spherical snow grain shape assumption was used in NorESM when other parts of the model were tuned originally."

**Comment:** *L357: NONSPH → SPH*

**Response and change:** Thanks for spotting this! It will be corrected.

**Comment:** *L516: reports*

**Response and change:** This will be corrected.

**Comment:** *L529: Perhaps: "at high latitudes up to XX K."*

**Response and change:** We will add: '"(up to $-4$ K in the extreme northeastern parts of Russia and locally $-7$ K in the Southern Ocean near Antarctica)".

**Comment:** *L531: Perhaps add to end of sentence something like "and therefore leads to large efficacy of the RF". This would better justify including the efficacy section in Section 7.*

**Response and change:** We will formulate this as "The climatic response is amplified by strong snow and sea ice feedbacks, which leads to a very high efficacy of the RF associated with changed snow grain shape."

**Comment:** *L552: Delete "to".*

**Response and change:** This will be corrected.

---

## Author Response (AR1)

Dear Prof. Alexander,

Point-by-point responses to the referee comments and short comments on the manuscript tc-2017-118 "Effects of snow grain shape on climate simulations: Sensitivity tests with the Norwegian Earth System Model" are given below. The line numbers refer to the marked-up version of the revised manuscript, where deletions are marked with red and additions with blue colour.

Sincerely, on behalf of myself and my coauthors,

Petri Räisänen

**Response to comments by Anonymous Referee #1**

We thank Anonymous Referee #1 for his/her constructive comments on the manuscript. Point-by-point responses to the comments are provided below. The referee comments are written in *italic* font, and our responses in normal font.

**Comment:** *Recommendation: Accept after minor revision. This is an interesting study which is suitable for publication in The Cryosphere. It is well-written.*

*lines 8-9. Say that the nonspherical grains are compared to spherical grains with the same specific surface area.*

**Response:** The effective radius is defined in our study using the ratio of volume to projected area rather than volume to total surface area. Therefore, even if the difference might in practice be small for snow, it is in principle more correct to use here "specific projected area" instead of "specific surface area".

**Change in the manuscript**: In the abstract, it is now said: "Therefore, for the same snow grain effective size (or equivalently, the same specific projected area), the snow broadband albedo is higher when assuming non-spherical rather than spherical snow grains, typically by 0.02–0.03." lines 10-11

**Comment:** *line 32. Cite also Dang et al. (2015)*

**Response and change:** We cite this paper in the revised manuscript. line 34

**Comment:** *line 47. If the snow grain contains concavities and hollows, then the projected area is not the appropriate measure, because internal surfaces also deflect photons. See Grenfell et al. (2005). Admittedly, although cavities are present in atmospheric ice crystals, they are uncommon in surface snow.*

**Response:** In fact, we agree only partially with this comment. It is certainly true that the single-scattering properties of non-spherical particles can be influenced by concavities and hollows, and it is also true that no definition of effective radius (including our Eq. 1) is *always* optimal. Hovever, it is not at all clear that defining the effective radius in terms of the total surface area $A$

$$r_{VA} = 3\frac{V}{A} \tag{1}$$

rather than in terms of the projected area $P$ (which is what we use in the manuscript, although with the notation $r_e$)

$$r_{VP} = \frac{3V}{4P} \tag{2}$$

would be an improvement, even in the case of concave particles. It should be noted that Grenfell et al. (2005) only consider $r_{VA}$ and do not compare its performance to $r_{VP}$, so their paper does not yield direct information on which choice actually works better. However, for concave particles $r_{VA} < r_{VP}$, and as noted by Grenfell et al. (2005) themselves, this leads to an overestimation of optical depth. This might not be a major issue for snow (for which the optical depth is usually very large), but in the case of cirrus clouds, it would necessarily lead to an underestimation of direct solar radiation. Furthermore, according to our (admittedly limited and unpublished) comparisons, $r_{VP}$ appears to be a better predictor of the single-scattering co-albedo of nonspherical particles than $r_{VA}$. That is, when plotted as a function of $r_{VP}$, the differences in co-albedo between different concave and convex particle shapes tend to be smaller than when plotted as a function of $r_{VA}$. The problems with using $r_{VA}$ to represent co-albedo can in fact also be seen from Fig. 3 of Grenfell et al. (2005). The values of co-albedo are systematically and even substantially higher for concave ice crystals than for ice spheres with the same $r_{VA}$. This is indeed what one would expect to see, because (i) for a given value of $r_{VA}$, $r_{VP}$ is larger for concave particles than for spheres and (ii) it is well known that the co-albedo generally increases with increasing particle size, when the particles are large compared to the wavelength.

Therefore, we adhere to our view that it is, in general, better to use the projected surface area rather than the total surface area in the definition of the effective size of nonspherical articles, but we also note that concavities and hollows do play a role.

**Change in the manuscript:** The following is stated in the revised manuscript: "While the SSPs of non-spherical particles (including snow grains) can be influenced by concavities and hollows (Grenfell et al. 2005), in general, the most relevant measure of their size for radiative transfer is the volume-to-projected area equivalent effective radius (e.g., Mitchell 2002) . . . " lines 53–55

**Comment:** *line 49, eq. 1. Point out that re is inversely proportional to specific surface area (SSA), a quantity that is commonly used in snow radiation work.*

**Response:** The $r_e$ is inversely proportional both to SSA and to the specific projected area (SPA). For completeness, it is perhaps best to add an equation showing these relationships.

**Change in the manuscript:** We note the following: "The $r_e$ is inversely proportional to the snow specific projected area (SPA; projected area per mass) and the specific surface area (SSA; total surface area per mass):

$$r_e = \frac{3}{4\rho_{\mathrm{ice}}\mathrm{SPA}} = \frac{3F}{\rho_{\mathrm{ice}}\mathrm{SSA}}, \tag{3}$$

where $\rho_{\mathrm{ice}}$ is the density of ice and the fluffiness parameter $F = \mathrm{SSA}/4\mathrm{SPA}$ (Grenfell et al. 2005) is $F = 1$ for convex particles such as spheres and $F > 1$ for concave particles." lines 58–63

**Comment:** *line 89. Change "retuning the snow grain size" to "increasing the snow grain size (of the nonspherical grains)"*

**Response and change:** This is reworded according to the suggestion. line 101

**Comment:** *line 96. "model model" is redundant.*

**Response and change:** This is corrected in the revised manuscript. line 108

**Comment:** *line 183. "abundant snow cover . . . in parts of Tibet . . . ". What does NorESM predict for snow cover and snow depth in Tibet? In reality, Tibetan snow is patchy and thin, with average depth peaking in February at only 2 cm (Flanner and Zender 2005, Figure 3b).*

**Response:** It is indeed true that NorESM overestimates the amount of snow in Tibet. The snow cover fraction in February is close to 80%, as compared with 30% in the NOAA SCE CDR (aka. Rutgers University) data, resulting in the distinct overestimate seen in Fig. 8a-b. The area-mean snow depth in February for the region considered by Flanner and Zender (2005) is about 25 cm in SPH and 31 cm in NONSPH. Incidentally, while overestimated snow cover likely exaggerates the "radiative forcing" associated with changed snow grain size, it is not obvious that overestimated snow depth works in the same direction. In fact, Fig. 1c of our manuscript indicates that snow grain shape has a larger effect on snow broadband albedo when the snow layer is relatively thin.

**Change in the manuscript:** We have added the following in Sect. 4.1 (as a footnote, to avoid the disruption of the flow of the main text): "The RF in Tibet may be exaggerated by NorESM's overestimation of snow cover in Tibet (see Fig. 8a–b below)." bottom of p. 7. In addition, the snow depth is mentioned in Sect. 4.3.1 where snow-related quantities are compared with observations. ". . . overestimation in Tibet (Fig. 8a–b), where snow depth is also overestimated (the February mean snow depth for the Tibetan Plateau region (30–40°N, 80–100°E) being 25 cm for SPH and 31 cm for NONSPH, as compared with roughly 10 cm for satellite microwave-derived data and only 2 cm for in situ data, see Fig. 3b in Flanner et al. (2005))". lines 310–312

**Comment:** *line 185-186. "in the southern parts of northern Eurasia . . . the change in snow albedo is largely masked by forests." This is also seen in a band of forest across North America at 50-60N between the Great Plains and the tundra.*

**Response and change:** This is true and we note it in the revised manuscript. lines 203–204

**Comment:** *line 209-210. Define "Q-flux".*

**Response:** The physical meaning of $Q$-fluxes is explained in connection with the description of the mixed-layer ocean model in Sect. 2 (lines 105–106 of the original manuscript and lines 117–118 of the marked-up manuscript): "The $Q$-flux (representing the implied horizontal and vertical heat flux into/out of the local mixed-layer column)...", so there is no need to repeat this explanation in Sect 4.2.

**Change in the manuscript:** For clarity, we have added a reference to Sect. 2 at the point where $Q$-fluxes are mentioned in Sect 4.2. "...especially because the $Q$-fluxes employed in the slab ocean model are based on a preindustrial simulation (see Sect. 2)". line 229

**Comment:** *line 296. Change "Figs. 8 and Fig. 9" to "Figs. 8 and 9".*

**Response and change:** This has been changed as suggested. line 322

**Comment:** *line 303. Change "or" to "of".*

**Response and change:** This has been corrected. line 329

**Comment:** *line 357. Change "NONSPH" TO "SPH". This is important.*

**Response and change:** Thanks for spotting this! It has been corrected. line 384

**Comment:** *line 372-373. "2 W m-2 in eastern Greenland (mainly due to BC)". This is probably excessive. The BC content at East Greenland AWS stations is only 2-4 ppb (Table 6 of Doherty et al. 2010).*

**Response:** We compared simulated surface-layer BC concentrations in the SPH experiment to the observations listed in Table 6 of Doherty et al. (2010). This comparison indicated a slight overestimation for the spring measurements (simulated BC concentrations 2–10 ppb, observed 2–7 ppb) and a more pronounced overestimation for the summer measurements (simulated BC concentrations 7–23 ppb, observed 1–20 ppb but mostly 1–4 ppb). However, it may be noted that the summer measurements are mostly at different sites than the spring measurements, and in fact no measurements are available in the region where the NorESM aerosol radiative effect is maximum. In that region, the simulated BC concentrations in surface snow in summer were as high as ∼30–40 ppb, which is very probably too much, even if some enrichment of BC in the surface snow is likely to happen during the snow melt season also in reality.

**Change in the manuscript:** In the interest of brevity, we have only added the following sentence regarding this issue in the revised manuscript: "The RE in Greenland may be excessive, as comparison with observed BC concentrations in Greenland (Table 6 in Doherty et al. 2010) suggested that NorESM likely overestimates the BC in surface snow especially in summer." lines 400–402

**Comment:** *line 441. Change "in lack of information" to "because of the lack of information".*

**Response and change:** This has been corrected as suggested. line 470

**Comments:** *line 447. "indistinguishable" is misspelled.*

*line 516. Change "report" to "reports".*

*line 603. Change "run" to "ran".*

**Response and change:** These typos have been corrected. lines 476, 547, 636

**Comment:** *Figure 3 caption line 3. Change "limit" to "threshold". Also on captions to Figures 5 and 6.*

**Response and change:** Yes, "threshold" is a better word here. This also applies to captions to Figs. 7 and 12. These have been corrected as suggested. Captions of Figs. 3, 5, 6, 7 and 12.

**Comment:** *Figure 7a. Give units on scale bar (probably micrometers).*

**Response and change:** We have added the units ($\mu$m) to the panel title of Fig. 7a (it is technically easier, and more consistent with our other figures).

**Comment:** *Figure 7b. A ratio (rather than percent-difference) might be esier for the reader to interpret. Also in Figure 11a.*

**Response and change:** We have modified these figures as suggested and updated the wording accordingly. lines 288–289, 381–383

**Comment:** *Figure 13 caption last line. These numbers will be easier to compare if they are given in the same units: "(24 ppb for hydrophilic BC and 8120 ppb for dust)".*

**Response and change:** This has been changed as suggested.

**Response to comments by Anonymous Referee #2**

We thank Anonymous Referee #2 for his/her constructive comments on the manuscript. Point-by-point responses to the comments are provided below. The referee comments are written in *italic* font, and our responses in normal font.

**Comment:** *The study investigates the impacts of including the effects of non-sphericity of snow grains in the NorESM model in slab ocean mode. They find that due to a smaller asymmetry parameter, snow broadband albedo is generally higher. Despite a global mean radiative forcing associated with this change of only -0.22 W/m2, this leads to considerable differences in the simulated equilibrium climate — particularly at high latitudes.*
*The paper is very well written, structured, illustrated and argued. It is, quite frankly, a pleasure to read and I have only extremely minor suggestions for improvement. It fits within the journal and I suggest to accept with only minor changes.*

*L109: Would it be relevant to mention (just briefly) if and how the sea ice model is dynamic and how this works on top of the slab ocean?*

**Response and change in the manuscript:** Yes, CICE4 is a dynamic sea ice model. We have added the following explanation: "Like fully coupled configurations, the slab ocean setup uses the full prognostic thermodynamic and dynamic configuration of the sea ice model. Ice velocities are prognostic and calculated based on winds from the atmosphere model and ocean currents specified from an earlier fully coupled run with NorESM (the same preindustrial control simulation as used for calculating the $Q$-fluxes for the slab ocean component). In addition to the thermodynamic response, this allows for transport and deformation of sea ice in response to changes in the atmospheric circulation." lines 122–126

**Comment:** *Eqs (3) and (4): Perhaps note that the time step DeltaT is multiplied*

*onto these rates to get the dr's used in Eqn (2).*

**Response and change:** In the revised manuscript, we have included $\Delta t$ on the rhs of Eqs. (3) and (4), so they directly give the $dr_{e,\mathrm{dry}}$ and $dr_{e,\mathrm{wet}}$ needed in Eq. (2). lines 151–157

**Comment:** *L226: at the end of the sentence, do you mean "SPH and ERA-Interim reach -7 K"?*

**Response:** No, the sentence is correct as it is. Its purpose is to stress that NON-SPH also differs a lot from SPH, not only ERA-Interim. (The maximum difference between SPH and ERA-Interim in Antarctica in summer is close to $-6\,\mathrm{K}$). Thus, no change in the manuscript. lines 245

**Comment:** *L246: Perhaps note that these diagnostic calculations were the same used to calculate the TOM RF.*

**Response and change:** This is noted in the revised manuscript. lines 265–266

**Comment:** *L264: "smaller contribution". It is not completely clear to me how you see that this contribution is smaller. Please explain better.*

**Response:** Quantifying precisely how much the change in snow grain size $r_e$ contributes to the albedo difference between NONSPH and SPH would require extensive and laborious off-line radiation calculations. However, even without performing such calculations, it is safe to say that the contribution from changed $r_e$ to the overall albedo diffence seen in Fig. 5b is (much) smaller than the contribution from increased snow and sea ice cover. The impact of the change in $r_e$ can best be discerned from other factors in regions with permanent snow cover (i.e., Greenland and Antarctica); it is also these regions which show the most consistent decrease in $r_e$ in the NONSPH experiment in Fig. 7b. By taking the difference between the actual albedo difference between NONSPH and SPH (Fig. 5b) and the corresponding albedo difference from diagnostic radiation calculations (Fig. 5a), we estimate that the reduced $r_e$ in NONSPH increases the snow albedo in these regions on average by 0.007, although the difference reaches 0.02 for some grid points. Albedo differences of this size are quite small compared to the differences between NONSPH and SPH in those regions where NONSPH features increased

snow and sea ice cover (which can be well above 0.1; Fig. 5b). Therefore, beyond reasonable doubt, the dominant contribution to the albedo differences in Fig. 5b comes from changed snow and sea ice cover rather than changed $r_e$.

**Change in the manuscript:** The revised manuscript includes a shorter version of the above explanation at the end of Sect. 4.3.: "The impact of this is most easily discernible in the permanently snow-covered regions of Antarctica and Greenland. In these regions, the actual albedo difference between NONSPH and SPH (Fig. 5b) is slightly larger than that derived from diagnostic radiation calculations (Fig. 5a); on average by 0.007 and by up to 0.02 for some grid points. These albedo changes are, however, much smaller than the albedo differences between NONSPH and SPH in regions of changed snow and sea ice cover." lines 294–299

**Comment:** *L322-330+Table 2: Is this part really necessary? If you want to shorten, this could be a place.*

**Response and change:** True enough, it is not strictly necessary. This part has been eliminated in the revised manuscript. lines 348–355

**Comment:** *L337: yes, because the default parameterization was used when other parts of the model were tuned originally.*

**Response and change:** The following sentence has been added to the revised manuscript: "In particular, the spherical snow grain shape assumption was used in NorESM when other parts of the model were tuned originally." lines 362–363

**Comment:** *L357: NONSPH → SPH*

**Response and change:** Thanks for spotting this! It has been corrected. line 384

**Comment:** *L516: reports*

**Response and change:** This has been corrected. line 547

**Comment:** *L529: Perhaps: "at high latitudes up to XX K."*

**Response and change:** We have added: '"(up to $-4$ K in the extreme northeastern parts of Russia and locally $-7$ K in the Southern Ocean near Antarctica)".

**Comment:** *L531: Perhaps add to end of sentence something like "and therefore leads to large efficacy of the RF". This would better justify including the efficacy section in Section 7.*

**Response and change:** We have formulated this as "The climatic response is amplified by strong snow and sea ice feedbacks, which leads to a very high efficacy of the RF associated with changed snow grain shape."

**Comment:** *L552: Delete "to".*

**Response and change:** This has been corrected.

**Response to comments by Dr. Cenlin He**

We thank Dr. Cenlin He for informing us about his very relevant recent article.

**Comment:** *The authors conducted a series of sensitivity simulations using the Norwegian Earth System Model to quantify the effects of snow grain shape, which could improve the understanding on the role of snow grain shape in climate modeling. For the authors information, a very recent study (He et al., 2017) did a detailed analysis and parameterization to account for snow grain shape effects on optical properties of both clean and dirty snow, which could be cited and discussed to improve the discussions in the manuscript.*

*Reference: He, C., Y. Takano, K. Liou, P. Yang, Q. Li, and F. Chen, 2017: Impact of C1 Snow Grain Shape and Black Carbon-Snow Internal Mixing on Snow Optical Properties: Parameterizations for Climate Models. J. Climate, 0, https://doi.org/10.1175/JCLI-D-17-0300.1*

**Response and change in the manuscript:** We have added two sentences about this work to the Introduction section of our paper: " Very recently, He et al. (2017b) developed another parameterization for the co-albedo and asymmetry parameter of snow for potential use in snow, land surface and climate models, based on

single-scattering calculations for spheres and three non-spherical shapes. This parameterization can be used for clean as well as dirty snow, as it includes the effects on co-albedo due to black carbon internally mixed with snow." Additionally, it is noted in Sect. 7.3 that for one of the shapes considered by He et al. (2017b), the value of asymmetry parameter appears to fit pretty well with the values derived by Ottaviani et al. (2015): "These values are, in fact, closer to the $g$ of large spherical snow grains ($g \approx 0.89$) or that of spheroids with an aspect ratio of 0.5 ($g \approx 0.86$; Fig. 4 in He et al., 2017b) than that of the OHC." lines 49–52, 508–509

**Other changes in the manuscript**

- For clarity, the name of the coauthor "Jens Boldingh Debernard" is now written as "Jens B. Debernard". "Boldingh" is a middle name, not a part of the surname.

- There is a slight modification concerning what is said about the the simulation of Arctic BC by CAM4-Oslo in Sect. 7.3. It is now stated that "Table 2 in Jiao et al. (2014) suggests that the BC concentrations in Arctic snow simulated by CAM4-Oslo ... are, overall, nearly unbiased compared to observations" (lines 514–517 in the marked-up manuscript). The mentioned table indicates, at face value, an underestimation of only 1–6%. The $\sim 20\%$ underestimate stated in the original manuscript was based on a slight misinterpretation of Fig. 1 in Jiao et al. (2014).

- A few typos and minor language errors have been spotted and corrected (e.g., there were a couple of instances where the text was referring to a wrong figure or wrong figure panels).

[revised manuscript text omitted]
^{\uparrow}_{\text{TOA}}$) with CERES EBAF-TOA Ed2.8 data (?) (Table ??). In NONSPH, $F^{\uparrow}_{\text{TOA}}$ is larger than in CERES EBAF by over 8 over Central Antarctica and by over 3 over Central Greenland, irrespective of whether all-sky or clear-sky fluxes are considered. The corresponding differences from CERES EBAF are substantially smaller for SPH. These differences are influenced by atmospheric absorption and scattering, and there are uncertainties both in the simulated cloud fields and in satellite cloud detection over snow surfaces (which could affect the CERES EBAF clear-sky fluxes). Even so, in principle, overestimated $F^{\uparrow}_{\text{TOA}}$ is consistent with the suggestion that snow albedo is overestimated in NONSPH.~~

**5 A tuning exercise**

Above, it has been shown that changing the snow grain shape assumption in NorESM has a substantial impact on the simulated climate. In general, the use of the OHC (i.e., non-spherical) shape assumption results in larger climatic biases than the use of the default spherical shape assumption, including a substantial cold bias at high latitudes (Fig. 4) and overestimated land snow cover and sea ice cover in the warm season (Figs. 8 and 9). While somewhat disappointing, this is not particularly surprising. Very frequently, even physically motivated parameterization changes lead to some deterioration of the simulate climate, if not accompanied by model retuning (e.g. Hourdin et al., 2017). In particular, the spherical snow grain shape assumption was used in NorESM when other parts of the model were tuned originally.

In this section, we explore one option for retuning: the reduction of snow albedo through adjusting the snow grain effective size $r_e$. This is motivated by Fig. 10  , which indicates that the NONSPH experiment probably overestimates snow albedo (at least in Greenland and Antarctica), and also by the fact that the snow grain $r_e$ is a relatively poorly known parameter. Here, we set a simple target for model retuning: we aim at reproducing the albedo simulated in the SPH experiment when using the OHC shape assumption for snow grains. Since snow albedo decreases with increasing snow grain size (Fig. 1c), this can be achieved by using a larger $r_e$ in connection with the OHC shape assumption. Based on diagnostic radiation calculations conducted for a three-year period of the SPH experiment, it was found that the global-mean difference between the OHC and spherical shape assumptions in both the surface and TOM net solar radiation is minimized, if for the OHC, the values of $r_e$ are multiplied by approximately 1.7.

Based on the above considerations, a retuned model version using the OHC was constructed with values of $r_e$ increased by ca. 70%. To achieve this, the limiting values of snow grain $r_e$ were increased by 70% from their default values: to $r_{e,\text{nonmelt}} = 850\,\mu\text{m}$ and $r_{e,\text{melt}} = 2550\,\mu\text{m}$ for snow on sea ice, and to $r_{e,0} = 92.7\,\mu\text{m}$, $r_{e,\text{rfrz}} = 1700\,\mu\text{m}$ and $r_{e,\text{rmax}} = 2550\,\mu\text{
[revised manuscript text omitted]